# EVA-MILP: Towards Standardized Evaluation of MILP Instance Generation

## Abstract

Mixed-Integer Linear Programming (MILP) is fundamental to solving complex decision-making problems. The proliferation of MILP instance generation methods, driven by machine learning's demand for diverse optimization datasets and the limitations of static benchmarks, has significantly outpaced standardized evaluation techniques. However, assessing the **fidelity** and **utility** of synthetic MILP instances remains a critical, multifaceted challenge. This paper introduces EVA-MILP – a comprehensive benchmark framework designed for the systematic and objective evaluation of MILP instance generation methods. Our framework provides a unified and extensible methodology, assessing instance quality across crucial dimensions: mathematical validity, structural similarity, computational hardness, and utility in downstream machine learning tasks. A key innovation is its in-depth analysis of **solver-internal features** – particularly by comparing distributions of key solver outputs including **root node gap**, **heuristic** success rates, and **cut plane** usage – leveraging the solver's dynamic solution behavior as an 'expert assessment' to reveal nuanced computational resemblances. By offering a structured approach with clearly defined solver-independent and solver-dependent metrics, our benchmark aims to facilitate robust comparisons among diverse generation techniques, spur the development of higher-quality instance generators, and ultimately enhance the reliability of research reliant on synthetic MILP data. The framework's effectiveness in systematically comparing the fidelity of instance sets is demonstrated using contemporary generative models. The code is available in https://github.com/iclr2026evamilp/EVA-MILP.

## 1 Introduction

Mixed-Integer Linear Programming (MILP) is a fundamental optimization framework extending standard Linear Programming (LP) by incorporating integer variables, enabling the modeling of discrete decisions and logical conditions often intractable with purely continuous models (Achterberg & Wunderling, 2013; Wolsey, 2020). Its versatility in capturing complex relationships like fixed costs, mutual exclusivity, and indivisible entities makes MILP essential for practical decision-making in fields such as supply chain optimization, scheduling, financial modeling, and network design (Hugos, 2018; Branke et al., 2015; Mansini et al., 2015; Al-Falahy & Alani, 2017).

Traditionally, MILP solver advancement relied on benchmark libraries like MIPLIB (Gleixner et al., 2021) and other real-world instance collections. *However, these static benchmarks often lack the scalability, diversity, or controllable characteristics vital for modern research.* These limitations are particularly acute with the rise of machine learning (ML) in optimization, where learning-based approaches for tasks like branching strategy selection or algorithm configuration demand large, varied datasets that often exceed available collections (Bengio et al., 2021). Consequently, a significant shift towards proactive **generation of MILP instances** has occurred (Bowly et al., 2020; Geng et al., 2023; Wang et al., 2023; Guo et al., 2024; Liu et al., 2024; Yang et al., 2024a; Zeng et al., 2024; Zhang et al., 2024). This trend towards instance generation is motivated by multiple highlighted advantages: fulfilling the need for extensive datasets in ML applications; achieving greater instance diversity than found in existing libraries (Gleixner et al., 2021); enabling control over computational difficulty for rigorous algorithm testing; facilitating the simulation of specific problem structures

(Liu et al., 2024); aiding solver testing and debugging (Gurobi Optimization LLC, 2021); and creating privacy-preserving surrogates for confidential real-world data.

Architecture of **EVA-MILP**:
a Standardized MILP Instance
Evaluation Framework

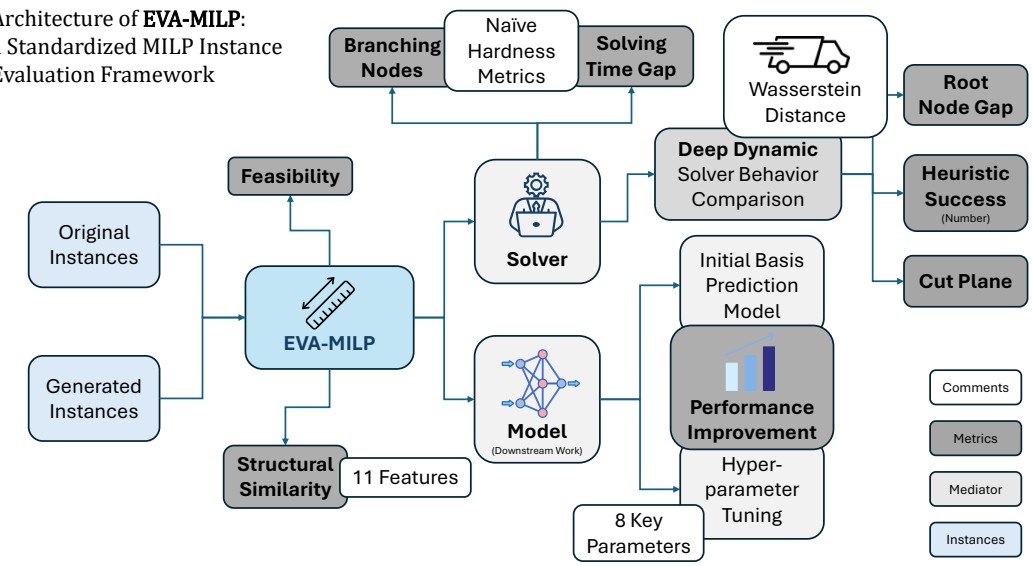

Figure 1: **Architecture of EVA-MILP**

Methodologies for MILP instance generation have evolved in tandem. Early approaches featured parameterized generators for **specific problem classes** such as the traveling salesman problems (Pilcher & Rardin, 1992; Vander Wiel & Sahinidis, 1995), set covering (Balas & Ho, 1980), or quadratic assignment problems (Culberson, 2002). Later techniques emphasized more **generalized feature-based descriptions** and **sampling** (Bowly et al., 2020; Smith-Miles & Bowly, 2015). Most recently, **deep learning based frameworks** like G2MILP (Geng et al., 2023), MILP-StuDio (Liu et al., 2024), and MILP-Evolve (Li et al., 2024) have gained prominence. These often represent MILP instances as graphs (e.g., bipartite graphs) and use **generative models** like variational autoencoders (Kipf & Welling, 2016) to learn and replicate specific structures from training data, *marking a significant potential advancement but also introducing new evaluation challenges.* More information about related work could be found in Appendix B

Despite this progress in generation, a critical and multifaceted challenge remains: the fair, objective, and comprehensive evaluation of synthetic MILP instance "quality". Assessing generator outputs effectively requires considering several dimensions: **(I)** *Ensuring fundamental solvability* (feasibility and boundedness) is crucial yet non-trivial (Wang et al., 2023; Geng et al., 2023; Liu et al., 2024). **(II)** *Generated instances should ideally mirror real-world structural patterns* (Gleixner et al., 2021), *but mere resemblance may not capture true computational behavior.* **(III)** *Instances must exhibit appropriate and controllable computational hardness*, typically evaluated via solver performance (Gurobi Optimization LLC, 2021), though achieving desired hardness levels consistently is a significant difficulty. Furthermore, *current evaluation protocols* (Geng et al., 2023; Bowly et al., 2020; Zhang et al., 2024) *often lack standardization, hindering robust comparisons and impeding progress.*

To address the lack of a standardized and comprehensive methodology for evaluating MILP instance generation techniques, this research introduces a novel benchmark framework. Our unified and extensible framework assesses instance quality across key dimensions – mathematical validity, structural similarity, computational hardness, solver interaction patterns, and utility for downstream ML tasks. By establishing an objective standard, we aim to facilitate fairer comparisons, guide the development of higher-quality generators, and improve the reliability of research utilizing synthetic MILP data. The **practical viability** and **robustness** of our framework is comfirmed by our tests on the super hard dataset with a strict 120-second time limit, proving *our solver-internal features are highly stable* and *confirming the approach is both low-cost and efficient.* Our experiments on open-source solvers SCIP and HiGHS further confirm that while *the framework is general, state-of-the-art solvers like Gurobi yield more precise internal features.* These findings not only validate our

framework but also highlight a limitation in current DL-based generation approaches. While representing MILP instances as bipartite graphs has enabled progress in achieving structural similarity, this representation may struggle to fully capture the intricate constraint relationships and inherent mathematical properties crucial for MILP feasibility and hardness. We posit that future advancements might benefit from shifting focus from direct graph structure manipulation towards methods that more directly model the underlying mathematical structure or solution space characteristics. The proposed benchmark provides the necessary tools to rigorously evaluate such novel approaches.

Our specific contributions are four-fold: **(I)** We integrate both **Solver-Independent Metrics** (covering fundamental properties and structural resemblance) and **Solver-Dependent Metrics** (including computational characteristics and downstream task utility) for a holistic assessment. **(II)** We incorporate detailed analysis of **solver-internal features** (e.g., **root node gap**, **heuristic** success counts, **cut plane** profiles), using the solver's behavior as an expert assessment to reveal deeper computational similarities. **(III)** The framework is designed for **extendability**, featuring a modular structure to easily accommodate new metrics, datasets, and generation techniques over time. **(IV)** We provide the community with clear, operational metrics and quantitative criteria to objectively benchmark diverse generator types (e.g., rule-based, statistical, and machine learning-based), thereby steering future development.

## 2 PRELIMINARIES

**MILP and its Bipartite Graph Representation**. We consider the standard MILP problem:

$$\min_{\mathbf{x} \in \mathbb{R}^n} \tilde{\mathbf{c}}^\top \mathbf{x}, \quad \textbf{s.t.} \quad \mathbf{A}\mathbf{x} \le \mathbf{b}, \ x_i \in \mathbb{Z}, \ \forall i \in \mathcal{I}.$$

Here, $\mathbf{x}$ is the vector of decision variables, $\tilde{\mathbf{c}}$ contains the objective coefficients, $\mathbf{A}$ is the constraint matrix, $\mathbf{b}$ is the vector of constraint bounds, and $\mathcal{I}$ identifies the integer variables.

Representing MILP instances as weighted bipartite graphs is an established practice in the relevant literature (Zhang et al., 2023; Gasse et al., 2019; Nair et al., 2021). In this common representation, each MILP instance corresponds to a graph $\mathcal{G}$ defined by three sets: constraint vertices $\mathcal{C}$, variable vertices $\mathcal{V}$, and edges $\mathcal{E}$ connecting them. The constraint partition $\mathcal{C} = \{c_1, \ldots, c_m\}$ includes one vertex $c_i$ for each of the $m$ constraints, where the vertex feature $c_i$ typically represents the bias term, i.e., $c_i = (b_i)$ (the $i$-th element of $\mathbf{b}$ from the MILP formulation). The variable partition $\mathcal{V} = \{v_1, \ldots, v_n\}$ contains one vertex $v_j$ for each of the $n$ variables, with the corresponding 9-dimensional feature vector $\mathbf{v}_j$ encoding information such as the objective coefficient $c_j$ (from the MILP vector $\mathbf{c}$), the variable type, and the bounds $l_j, u_j$ (from the MILP vectors $\mathbf{l}$ and $\mathbf{u}$). Edges in $\mathcal{E}$ only exist between vertices from different partitions ($c_i \in \mathcal{C}$ and $v_j \in \mathcal{V}$). An edge $e_{i,j}$ connects $c_i$ and $v_j$ if the corresponding element $A_{i,j}$ in the constraint matrix $\mathbf{A}$ is non-zero; its associated feature $e_{i,j}$ is described by this coefficient, i.e., $e_{i,j} = (A_{i,j})$. If $A_{i,j}$ is zero, the edge $e_{i,j}$ is considered absent.

**Linear Programming Relaxation.** Given the MILP problem:

$$z_{\text{MILP}} = \min_{\mathbf{x}} \{\tilde{\mathbf{c}}^\top \mathbf{x} \mid \mathbf{A}\mathbf{x} \le \mathbf{b}, x_i \in \mathbb{Z} \ \forall i \in \mathcal{I} \subseteq \{1, \ldots, n\}\}.$$

The *LP relaxation* is obtained by omitting the integrality constraints:

$$z_{\text{LP}} = \min_{\mathbf{x}} \{\tilde{\mathbf{c}}^\top \mathbf{x} \mid \mathbf{A}\mathbf{x} \le \mathbf{b}\}.$$

Let $\mathcal{P} = \{\mathbf{x} \in \mathbb{R}^n \mid \mathbf{A}\mathbf{x} \le \mathbf{b}\}$ be the feasible region of the LP relaxation. The optimal objective value of the LP relaxation provides a lower bound (for minimization problems) to the optimal objective value of the original MILP, i.e., $z_{\text{LP}} \le z_{\text{MILP}}$.

**Duality Gap.** In the context of solving the MILP $\min \tilde{\mathbf{c}}^\top \mathbf{x}$, the *duality gap* measures the difference between the objective value of the best known feasible integer solution, $z_{\text{incumbent}}$ (primal bound or upper bound, UB), and the best proven objective bound, $z_{\text{bound}}$ (dual bound or lower bound, LB), typically derived from LP relaxations ($z_{\text{bound}} \ge z_{\text{LP}}$). The absolute gap is $G_{\text{abs}} = z_{\text{incumbent}} - z_{\text{bound}}$ (assuming $z_{\text{incumbent}} \ge z_{\text{bound}}$). The relative gap, often used as a termination criterion, quantifies the remaining gap relative to one of the bounds, e.g.:

$$G_{\text{rel}} = \frac{z_{\text{incumbent}} - z_{\text{bound}}}{\max(1, |z_{\text{incumbent}}|)}.$$

Optimality is proven when $z_{\text{incumbent}} = z_{\text{bound}}$, or $G_{\text{abs}}$ (or $G_{\text{rel}}$) is within a predefined tolerance $\epsilon \geq 0$.

Additional preliminaries can be found in Appendix E.

## 3 METHODOLOGY

This section outlines the methodology employed to evaluate the quality and characteristics of generated MILP instances. We compare instances generated using reproductions of three open-source methods – G2MILP (Geng et al., 2023), ACM-MILP (Guo et al., 2024), and DIG-MILP (Wang et al., 2023) – against a baseline set of 'original' instances. This baseline set, comprising Set Cover, Combinatorial Auction (CA), Capacitated Facility Location (CFL), and Independent Set (IS) problems, was synthesized using the Ecole library (Prouvost et al., 2020) (the specific parameter settings used for this synthesis are detailed in the Appendix A.1), while IP and LB come from two challenging real-world problem families used in ML4CO 2021 competition (Gasse et al., 2022). To capture different facets of similarity and fidelity between the generated and original sets, we utilize a suite of metrics. These metrics are organized into two primary categories based on their reliance on the underlying optimization solver: **(I) Solver-Independent Metrics**, which assess inherent properties of the MILP instances themselves (e.g., feasibility, structure) without regard to any specific solution process; and **(II) Solver-Dependent Metrics**, which evaluate aspects intrinsically linked to the behavior and performance of a particular solver (primarily Gurobi) when applied to the instances. This dual approach allows for a robust assessment, considering both fundamental instance properties and their practical implications for optimization algorithms (See the whole framework in Figure 1).

### 3.1 DATASETS

This study evaluates MILP instances from two sources: original instances (baselines and validation) and those from three generative models (G2MILP, ACM-MILP, DIG-MILP). Baselines include four synthetic datasets (SC, CA, CFL, IS) from Ecole (Prouvost et al., 2020) (details in Appendix A.1) and public benchmarks from the ML4CO Competition 2024 (Gasse et al., 2022) for pipeline validation. *Due to the inherent specialization of generative models to particular problem types*, our comparative analysis, which forms the core of our evaluation, specifically examines SC instances from G2MILP, CA instances from ACM-MILP and DIG-MILP, and IS instances from ACM-MILP and G2MILP to demonstrate our metrics. The methodology's applicability to other datasets and models has been confirmed (see more in Section 4.1)

### 3.2 SOLVER-INDEPENDENT METRICS

#### 3.2.1 FEASIBILITY RATIO

This metric measures the proportion of instances that are both **feasible** and **bounded**. Most methods maintain near-perfect feasibility across datasets, with only minor exceptions (e.g., G2MILP on MIS at $\eta = 0.10$). Detailed results are given in Table 10 in the Appendix.

#### 3.2.2 STRUCTURAL SIMILARITY

To quantitatively assess the **structural similarity** between two sets of MILP instances, following (Geng et al., 2023), we implemented a feature-based comparison method. Each instance is first converted into a graph representation, from which a vector of **11 predefined structural features** is extracted. For each feature, we compute the **Jensen–Shannon Divergence (JSD)** between the distributions of the two sets and transform it into a similarity score

Table 1: **Structural Similarity**

| Problem | Model | Similarity at Masked Ratio ($\eta$) | | | |
|---|---|---|---|---|---|
| | | 0.01 | 0.05 | 0.10 | 0.20 |
| CA | ACM-MILP | — | 0.889 | 0.867 | 0.892 |
| CA | DIG-MILP | 0.861 | 0.860 | 0.879 | — |
| IS | ACM-MILP | — | 0.734 | 0.731 | 0.699 |
| IS | G2MILP | 0.486 | 0.473 | 0.466 | — |
| SC | G2MILP | **0.990** | **0.950** | **0.921** | — |

$S_i = 1 - \frac{JSD_i}{\log(2)}$, where a score of 1 indicates identical distributions and 0 indicates maximal divergence. The overall structural similarity between the two sets is the arithmetic mean of these feature

scores. Overall, the JSD-based metric (Table 1) indicates that ACM-MILP and DIG-MILP closely match the structural characteristics of CA data (around 0.86–0.89), while ACM-MILP achieves moderate similarity on IS (around 0.70) and G2MILP is lower (around 0.47). For Set Cover, G2MILP performs best, reaching very high similarity (around 0.92–0.99), especially at lower mask ratios.

## 3.3 SOLVER-DEPENDENT METRICS

### 3.3.1 BRANCHING NODES

Computational hardness is measured by the number of **branch-and-bound nodes** explored by Gurobi, using the relative error $\left| \frac{\sum Nodes_{\text{gen}} - \sum Nodes_{\text{train}}}{\sum Nodes_{\text{train}}} \right| \times 100\%$. As shown in Table 11 (Appendix F), hardness varies strongly by model and problem type: G2MILP on IS can exceed 50,000 % RE and often times out, ACM-MILP on CA reaches over 26,000 % RE, while DIG-MILP and other settings remain closer to baseline.

### 3.3.2 SOLVING TIME GAP

This metric measures the percentage difference in average solving times between generated and original instances, where smaller gaps indicate closer solver-conditional hardness. Table 13 in Appendix shows that G2MILP for SC keeps gaps modest ($\approx$11–22%), DIG-MILP for CA remains near baseline ($\approx$ 20–29%), while ACM-MILP can greatly alter hardness (e.g., 2400%+ increase for CA or $\sim$47–49% faster for IS).

### 3.3.3 HYPERPARAMETER TUNING

Automatically optimizing solver hyperparameters is known to be crucial for achieving peak performance on complex algorithms like MILP solvers (Hutter et al., 2011). Therefore, we employed the Sequential Model-based Algorithm Configuration (**SMAC3**) framework (Lindauer et al., 2022), which utilizes Bayesian optimization. Following the approach in (Liu et al., 2024), the tuning process targeted **8 key Gurobi parameters**. These parameters were selected as they govern diverse and fundamental components of

Table 2: **Solving time improvement after Gurobi hyperparameter tuning**. Compares the average wall-clock time (seconds) on the test set for the default configuration versus the best configuration found by SMAC3

| Dataset | Source | $\eta$ | Default(s) | Best(s) | Improv. (%) |
|---|---|---|---|---|---|
| IS | ACM-MILP | 0.05 | 0.339 | 0.370 | -9.17 |
| | | 0.1 | 0.351 | 0.280 | 20.25 |
| | | 0.2 | 0.339 | 0.377 | -11.13 |
| | Original | — | 0.336 | 0.366 | 8.98 |
| CA | DIG-MILP | 0.01 | 0.107 | 0.041 | **61.35** |
| | | 0.05 | 0.109 | 0.042 | **61.63** |
| | | 0.1 | 0.108 | 0.042 | **61.38** |
| | Original | — | 0.107 | 0.042 | **61.26** |
| SC | G2MILP | 0.01 | 0.211 | 0.118 | 43.97 |
| | | 0.05 | 0.211 | 0.118 | 44.06 |
| | Original | — | 0.229 | 0.221 | 3.31 |

the MILP solution process, including primal heuristics, search strategy focus, branching decisions, presolving routines, cutting plane generation, and node LP solution methods. The significant impact of these core solver components, and thus the parameters controlling them, on overall performance is well-documented in the MILP literature (e.g., Lodi & Zarpellon, 2017; Achterberg, 2009; Berthold, 2012). The objective function for SMAC3 was the minimization of the mean wall-clock solve time across the instances within a designated tuning set. The primary goal of this hyperparameter tuning experiment was to evaluate the **generalization** capability of the optimized Gurobi configurations using generated MILP instances. SMAC3 tuning (Table 2) gives model- and dataset-specific gains. ACM-MILP (IS) decreases about **10%** at $\eta = 0.1$. DIG-MILP (CA) consistently cuts time by about **61%**, and G2MILP (SC) by about **44%**, while the original SC set improves only about **3%**.

### 3.3.4 INITIAL BASIS PREDICTION

Following the methodology of (Fan et al., 2023), we investigated predicting an **initial basis** for MILP relaxations using a GNN. While adhering to the principles outlined by Fan et al., our implementation was developed independently from scratch. For this purpose, a GNN model was trained specifically on generated MILP instances to predict the basis status (basic or non-basic at lower/upper bounds)

for variables and slack variables associated with the MILP relaxation. The GNN's predictions were subsequently refined into a numerically stable basis using established techniques. To evaluate the effectiveness of training on generated data versus original data for this task, *we measured the impact of using the initial basis predicted by the GNN on Gurobi's performance metrics (solving time, runtime). This was compared against Gurobi's default setting and potentially a model trained on original data, using unseen test instances.* The validity and effectiveness of our re-implemented approach are substantiated by the experimental outcomes. Resuls are in Table 15 (Appendix K)

### 3.3.5 SOLVER-INTERNAL FEATURES

We introduce a novel, computationally efficient methodology to evaluate MILP instance fidelity using **solver-internal features**: metrics from Gurobi's deterministic solving process. We first validated the solver-internal–feature methodology via a split-half experiment (see Appendix L.1). Figure 2 shows one representative result, where random original-set subsets exhibit very low **1-Wasserstein distances** (e.g. 0.130 for heuristic, 0.188 for root-node gap, 0.232 for cut-plane profiles), confirming its ability to capture authentic inter-set similarity. To further demonstrate practical viability, we applied the method to the **hard** ML4CO *item_placement* dataset under a strict 120-second limit (see more in Section 4.1. This shows that we do not need to fully solve the instance to extract the features. Solver-internal features extracted from Gurobi proved stable and inexpensive to compute. What's more, comparison with experiments of open-source solvers **SCIP** and **HiGHS** (see more in Section 4.2) indicates that, although the framework is general, state-of-the-art solvers like Gurobi provide more precise internal metrics.

Table 3: **Comparison of Root Node Gap Distributions**

| Problem | Model | $\eta$ | Mean Gap (%) Gen | Bench | Std Dev (%) Gen | Bench | $W_1$ Dist. |
|---------|-------|--------|------|-------|------|-------|-------------|
| CA | DIG-MILP | 0.01 | 1.11 | 2.47 | 1.22 | 2.60 | 1.3001 |
|    |          | 0.05 | 1.09 | 2.47 | 1.17 | 2.60 | 1.3787 |
|    |          | 0.10 | 1.18 | 2.47 | 1.26 | 2.60 | 1.2982 |
|    | ACM-MILP | 0.05 | 7.93 | 2.47 | 3.85 | 2.60 | **5.4638** |
|    |          | 0.10 | 7.49 | 2.47 | 3.60 | 2.60 | 5.0187 |
|    |          | 0.20 | 9.54 | 2.47 | 4.40 | 2.60 | **7.0666** |
| IS | ACM-MILP | 0.05 | 2.65 | 3.42 | 1.82 | 1.34 | 0.8818 |
|    |          | 0.10 | 2.66 | 3.42 | 1.81 | 1.34 | 0.8646 |
|    |          | 0.20 | 2.45 | 3.42 | 1.80 | 1.34 | 1.0388 |
| SC | G2MILP | 0.01 | 8.25 | 7.61 | 4.19 | 4.12 | 0.6690 |
|    |          | 0.05 | 7.75 | 7.61 | 4.27 | 4.12 | **0.2294** |
|    |          | 0.10 | 8.05 | 7.61 | 4.12 | 4.12 | **0.5103** |

Table 4: **Comparison of Heuristic Success Frequency Distributions**

| Problem | Model | $\eta$ | Gen. Stats Mean | Std | Base Stats Mean | Std | $W_1$ Dist. |
|---------|-------|--------|------|-----|------|-----|-------------|
| CA | ACM-MILP | 0.05 | 2.549 | 0.500 | 2.115 | 0.379 | 0.434 |
|    |          | 0.10 | 2.552 | 0.501 | 2.115 | 0.379 | 0.437 |
|    |          | 0.20 | 2.457 | 0.500 | 2.115 | 0.379 | 0.342 |
|    | DIG-MILP | 0.01 | 2.238 | 0.451 | 2.115 | 0.379 | 0.135 |
|    |          | 0.05 | 2.241 | 0.439 | 2.115 | 0.379 | 0.126 |
|    |          | 0.10 | 2.258 | 0.440 | 2.115 | 0.379 | 0.143 |
| IS | ACM-MILP | 0.05 | 1.662 | 0.473 | 18.883 | 12.112 | 17.221 |
|    |          | 0.10 | 1.746 | 0.435 | 18.883 | 12.112 | 17.137 |
|    |          | 0.20 | 1.849 | 0.358 | 18.883 | 12.112 | 17.034 |
| SC | G2MILP | 0.01 | 2.619 | 0.533 | 2.702 | 0.491 | **0.083** |
|    |          | 0.05 | 2.600 | 0.548 | 2.702 | 0.491 | **0.102** |
|    |          | 0.10 | 2.598 | 0.541 | 2.702 | 0.491 | **0.104** |

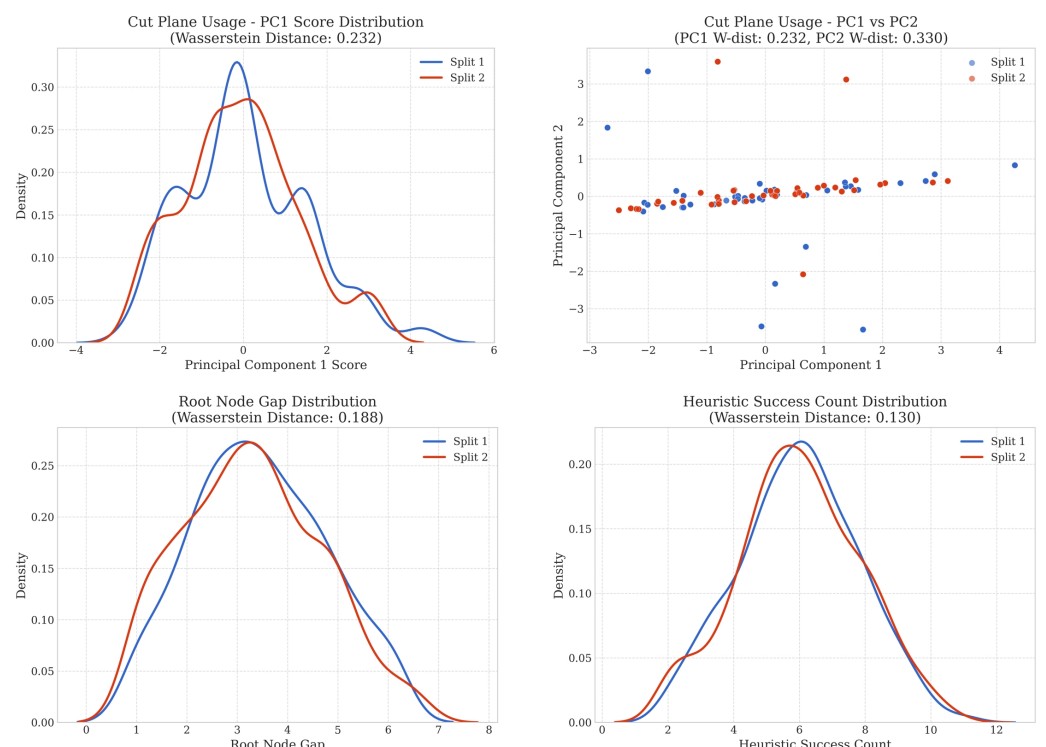

Figure 2: **Internal Cut-plane Comparison of IS (Demonstration of Identical Distribution)**

With fixed Gurobi settings, we extracted: **Root Node Gap**, **Heuristic Success Count**, and **Cut Plane Usage vectors**. Distributions for Root Node Gap and Heuristic Success Count were compared using the 1-Wasserstein distance ($W_1$). For Cut Plane Usage, per-instance vectors were preprocessed, underwent PCA, and distributions of scores on dominant principal components were then compared using the $W_1$ distance.

**Results** Table 3 presents $W_1$ distances for root gap distributions (lower $W_1$ indicates higher similarity). G2MILP highly replicated SC root gaps (e.g., $W_1 = 0.2294$ at $\eta = 0.05$). In contrast, DIG-MILP and ACM-MILP significantly diverged for CA. ACM-MILP showed moderate success for IS at lower $\eta$, diminishing as $\eta$ increased.

For primal heuristic success distributions (Table 4, $W_1$ distance), DIG-MILP (e.g., $W_1 \approx 0.14$) and ACM-MILP (e.g., $W_1 \approx 0.34 - 0.44$) achieved good similarity for CA problems. G2MILP (SC) demonstrated remarkable similarity ($W_1 \approx 0.08 - 0.10$). Conversely, ACM-MILP (IS) instances diverged significantly ($W_1 \approx 17.0 - 17.2$).

Table 5: **Comparison of Cutting Plane Usage Distributions**

| Problem | Model | $\eta$ | Wasserstein Distance | | |
| --- | --- | --- | --- | --- | --- |
| | | | PC1 | PC2 | PC3 |
| CA | ACM-MILP | 0.05 | 1.5570 | 0.8352 | 0.5349 |
| | | 0.10 | 1.4837 | 0.7155 | 0.3570 |
| | | 0.20 | 1.6889 | 0.3166 | 0.3731 |
| | DIG-MILP | 0.01 | 1.4711 | 0.6673 | 0.3601 |
| | | 0.05 | 1.4868 | 0.6826 | 0.3047 |
| | | 0.10 | 1.5392 | 0.4394 | 0.5988 |
| IS | ACM-MILP | 0.05 | 0.8944 | 0.4986 | 0.2517 |
| | | 0.10 | 0.9354 | 0.3942 | 0.0963 |
| | | 0.20 | 1.0097 | 0.5735 | 0.1199 |
| SC | G2MILP | 0.01 | **0.0707** | 0.1271 | 0.1124 |
| | | 0.05 | **0.1714** | 0.1425 | 0.0677 |
| | | 0.10 | **0.1374** | 0.1028 | 0.1375 |

For cutting plane usage (Table 5), G2MILP (SC) achieved the highest similarity (e.g., PC1 $W_1 \approx 0.07 - 0.17$). In contrast, ACM-MILP and DIG-MILP for CA instances showed significant divergences (DIG-MILP PC1 $W_1 \approx 1.47 - 1.69$). ACM-MILP (IS) demonstrated intermediate similarity (PC1 $W_1 \approx 0.89 - 1.01$).

## 4 EXPERIMENTS

### 4.1 EFFICIENCY TEST ON SUPER HARD INSTANCES

To validate the practical feasibility and robustness of our solver-internal feature analysis method, we designed an efficiency test. The core goal of this experiment is to demonstrate that even under strict computational constraints (i.e., tight time limits) and when facing a well-known set of hard instances, our method can still stably and efficiently extract meaningful feature distributions. This supports our key hypothesis that obtaining such deep behavioral features does not require solving instances to optimality, thereby making the evaluation both low-cost and highly efficient.

**Dataset Selection**: We used the item placement dataset from the ML4CO competition, which is recognized for its high solving difficulty. A total of 1000 instances were selected for testing.

**Computational Budget Constraint**: To simulate scenarios with limited computational resources, we imposed a strict time limit of **120 seconds** for the Gurobi solver on each instance.

**Validation Method**: We employed a split-half validation approach to evaluate the stability of

Table 6: **W-1 distances of solver-internal features across item placement halves**

| Feature | W-1 Dist. |
|---|---|
| Root Node Gap | 0.7613 |
| Heuristic Success Count | 0.7320 |
| Cut Plane Usage | |
| – PC1 | 0.0917 |
| – PC2 | 0.0870 |
| – PC3 | 0.0904 |

the extracted features. The 1000 solver logs were randomly divided into two halves of 500 instances each. We then independently extracted solver-internal features—including Root Node Gap, Heuristic Success Count, and Cut Plane Usage—from both halves and computed the distributional similarity between them. Strong stability and reliability of the extraction method are indicated if these random subsets, originating from the same underlying distribution, exhibit highly similar feature distributions (i.e., a low Wasserstein distance).

The split-half validation demonstrates remarkable stability in the extracted solver-internal features, even under the strict 120-second time limit. As shown in Table 6, the Cut Plane Usage metric exhibits exceptionally low 1-Wasserstein distances for its top three principal components (PC1: 0.0917, PC2: 0.0870, and PC3: 0.0904). These near-zero values indicate that the solver's strategic application of cuts—a crucial early-stage behavior—is highly consistent across the two random data subsets. Similarly, the Root Node Gap (0.7613) and Heuristic Success Count (0.7320) show strong distributional similarity, confirming that a stable signal for these metrics can also be captured long before the solver reaches optimality. Collectively, these results validate that a truncated solving process is sufficient to extract a reliable and consistent "fingerprint" of solver behavior, confirming the low-cost and efficient nature of our methodology.

**Conclusion:** This experiment strongly demonstrates the practical value of our framework: ① **High Efficiency and Low Cost**: The results confirm that a truncated solving process (120-second timeout) is sufficient to extract stable and representative solver-internal feature distributions. This substantially reduces the time and computational cost of evaluating large-scale or extremely hard instance sets. ② **Feature Stability**: The very small Wasserstein distances observed for Root Node Gap and Cut Plane Usage indicate that these early-stage solver behavior features are highly stable. ③ **Practical Feasibility**: The success of this test shows that the EVA-MILP framework is not only theoretically sound but also practically applicable for evaluating MILP instances that are difficult to solve to optimality within a reasonable time frame and that reflect real-world complexity.

### 4.2 CROSS-SOLVER COMPARISON AND STABILITY ANALYSIS

To assess the generality of the EVA-MILP framework and probe the internal stability of different solvers, we extended our evaluation to include the open-source solvers SCIP and

Table 7: **1-Wasserstein distances across solvers on IS split-half validation**

| Feature | Gurobi (Baseline) | SCIP | HiGHS |
|---|---|---|---|
| Root Node Gap | **0.1884** | 1.7530 | 26.7297 |
| Heuristic (PC1) | **0.1300** | 0.3190 | 0.2368 |
| Cut Plane (PC1) | 0.2318 | 0.7000 | **0.0612** |

HiGHS. We conducted the same IS split-half validation procedure, using Gurobi as the stable baseline for comparison. Table 7 summarizes the key results.

The data reveals a complex and multi-faceted picture of solver stability. First, examining the **Root Node Gap**, we observe dramatic differences. Gurobi exhibits exceptional stability with a Wasserstein distance of just **0.1884**. In stark contrast, HiGHS shows a distance of 26.7297, over **140 times** larger than Gurobi's, signifying severe instability in its root node analysis. This suggests that the initial problem assessment and relaxation strategies in HiGHS are profoundly sensitive to small changes in problem structure. Second, the results for the primary principal component of **Heuristic** features show a similar trend, though less pronounced. Gurobi remains the most stable (0.1300), while SCIP and HiGHS are approximately **2.5x** and **1.8x** less stable, respectively. This implies Gurobi's primal heuristic strategies are more robust. Interestingly, the analysis of **Cut Plane** features presents a contrasting result. Here, HiGHS is by far the most stable solver (0.0612), with a consistency nearly 4 times greater than Gurobi's (0.2318). This suggests that HiGHS may employ a more deterministic or conservative cut generation strategy, whereas Gurobi and SCIP might use more adaptive, opportunistic approaches that are inherently more variable.

From this detailed analysis, we draw two main conclusions. First, the successful application to SCIP and HiGHS validates that *the EVA-MILP framework is indeed solver-agnostic and generalizable*. Second, and more importantly, the quality of instance evaluation is intrinsically linked to solver stability, which is not monolithic but *highly feature-dependent*. Gurobi's superior stability in the critical root-node phase makes it the most reliable choice for a baseline. The extreme volatility of the root node gap in HiGHS, for instance, could obscure subtle computational similarities that a more stable solver like Gurobi can reliably detect.

In summary, this cross-solver comparison highlights a powerful secondary application of our framework. Beyond its primary purpose of instance evaluation, *EVA-MILP serves as an effective diagnostic tool for profiling the internal dynamics of MILP solvers*, offering quantitative insights into their stability and behavioral responses to perturbations across different strategic components.

## 5 DISCUSSION

**Finding 1:** *Superficial structural similarity is an unreliable predictor of computational behavior and difficulty in generated MILP instances.* ACM-MILP (CA) instances showed high structural similarity but were exceptionally difficult to solve, with Root Node Gap discrepancies suggesting that structural metrics miss key complexity drivers or that models introduce subtle, difficulty-enhancing variations.

**Finding 2:** *Simple outcome metrics, such as solving time or branching nodes, do not adequately reveal the source of problem difficulty or underlying changes to instance structure.* This is exemplified by ACM-MILP (IS) instances, which, despite some similar outcome metrics (e.g., solving time), exhibited vastly different heuristic behavior, indicating altered internal structures that impact solvability in ways not captured by these surface-level performance measures.

**Finding 3:** *The effectiveness of generative models is problem-dependent, linked to how well their modification strategies align with the core structural determinants of hardness for specific problem types.* G2MILP's difficulty in generating realistic IS instances is likely due to IS problem hardness being rooted in global graph topology (Tarjan & Trojanowski, 1977; West, 2001), which its local edge constraint modifications struggle to preserve. G2MILP's better performance on SC problems suggests its direct manipulation of constraint configurations (Balas & Ho, 1980) more effectively captures their structural hardness.

## 6 CONCLUSION

We introduce EVA-MILP , a comprehensive benchmark framework designed to address the critical shortcomings of traditional evaluation methods for generated MILP instances. Our findings reveal that metrics based on superficial structural similarity or simple outcomes often fail to capture an instance's true computational nature and difficulty. EVA-MILP facilitates a more accurate and holistic assessment by shifting the focus towards solver-perceived difficulty and the practical utility of instances in downstream applications. By enabling this multifaceted methodology, our framework will be instrumental in guiding the development of higher-quality instance generators and fostering impactful research within the field of mathematical optimization.

## 7 ETHICS STATEMENT

The authors adhere to the ICLR Code of Ethics. This research introduces a methodological framework using only public and synthetic data, involves no human subjects, and supports privacy-preserving research. As an evaluation tool, the framework has a low risk of direct misuse. Potential research pitfalls, such as high computational cost or a narrow focus on metrics, are actively mitigated by the framework's demonstrated efficiency and extensible design. Our goal is to enhance the transparency and rigor of research in the optimization community.

## 8 REPRODUCIBILITY STATEMENT

We are committed to ensuring the reproducibility of our research. All datasets used in our experiments are either publicly available (ML4CO Competition) or were generated using the open-source Ecole library, with detailed generation parameters provided in Appendix A. Our experimental environment, including hardware and software versions, is detailed in Appendix D. Key hyperparameters for all experiments are summarized in the tables in Appendix C. The detailed methodology for each evaluation metric is fully described in Section 3 and its corresponding appendices G. The code is available in `https://github.com/iclr2026evamilp/EVA-MILP`. You can fully reproduce all experiments and results through the provided codebase.

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

APPENDIX CONTENTS

## A DATASETS

### A.1 ECOLE SYNTHESIZED INSTANCES

To facilitate various experiments, including model training and evaluation across different metrics, 4 synthetic datasets of MILP instances was generated. This process utilized the **Ecole** library (Prouvost et al., 2020), a platform designed for machine learning research in combinatorial optimization.

**Parameter Randomization:** Parameters like the number of constraints, variables, and density are randomly sampled for each instance within the ranges specified in the configuration file. This ensures diversity in the generated dataset. The details of synthesizing parameters are in Table 8.

Table 8: Parameter Settings for MILP Instance Synthesis by Problem Type

| Problem Type | Parameter | Value / Range |
|---|---|---|
| Set Cover (SC) | Constraints Range | [200, 800] |
| | Variables Range | [400, 1600] |
| | Density Range | [0.05, 0.2] |
| Capacitated Facility Location (CFL)* | Ratio (Facilities/Customers) | 0.5 |
| | Constraints Range | [50, 150] |
| | Variables Range | [500, 5000] |
| | Density Range | [0.01, 0.3] |
| Combinatorial Auction (CA)** | Number of Items (n_items) | Auto-calculated (null) |
| | Value Range (min/max) | [1, 100] |
| | Constraints Range | [50, 200] |
| | Variables Range | [80, 600] |
| | Density Range | [0.02, 0.1] |
| Independent Set (IS)*** | Variables (Nodes) Range | [480, 520] |
| | Density (Edge Prob.) Range | [0.01, 0.015] |
| | Constraints Range | N/A (Derived) |

*For CFL, the input 'constraints' and 'variables' ranges influence the number of customers and facilities, which in turn determine the actual model size. The 'density' range specified might not be directly used by the 'ecole' generator for CFL.

**For CA, the input 'constraints', 'variables', and 'density' ranges influence the number of bids and potentially items (if 'n_items' is null), which determine the actual model size.

***For IS, the 'variables' range directly corresponds to the number of nodes. The 'constraints' range is implicitly determined by the number of nodes and the edge probability (density) and is not directly set via 'min/max_constraints' in the config.

### A.2 COMPATIBILITY TESTING WITH PUBLIC DATASETS (NIPS ML4CO)

In addition to utilizing the synthetically generated dataset described above, the compatibility and robustness of the various analysis pipelines and tools developed (e.g., for feature extraction, solver log parsing, model evaluation) were verified using publicly available MILP benchmark instances. Specifically, datasets from the **ML4CO Competition** (Gasse et al., 2022) were employed for testing purposes. Successfully processing and running analyses on these standard benchmarks demonstrated the broader applicability and compatibility of the developed experimental framework beyond the specific synthetically generated instances. More results are in Appendix L.3

## B MORE RELATED WORK

The generation of high-quality MILP instances is crucial for developing, testing, and tuning both traditional solvers and modern ML approaches for combinatorial optimization. However, the scarcity of diverse, representative real-world instances, often due to proprietary constraints or collection difficulties, presents a significant bottleneck. This has spurred research into synthetic instance generation techniques. Recently, deep learning (DL) has emerged as a promising direction for MILP instance generation, aiming to automatically learn complex data distributions and structural features from existing instances without requiring explicit, expert-designed formulations. Several DL-based frameworks have been proposed:

**VAE-based Approaches:** G2MILP (Geng et al., 2023) introduced the first DL-based framework, using a masked Variational Autoencoder (VAE) paradigm. It iteratively corrupts and replaces constraint nodes in the bipartite graph representation. While capable of generating instances structurally similar to the training data, G2MILP does not explicitly guarantee the feasibility or boundedness of the generated instances, potentially leading to unusable samples for certain downstream tasks. DIG-MILP (Wang et al., 2023), also VAE-based, addresses the feasibility and boundedness guarantee by leveraging MILP duality theories and sampling from a space including feasible solutions for both primal and dual formats. However, this can come at the cost of structural similarity compared to the original data. ACM-MILP (Guo et al., 2024) refines the VAE approach by incorporating adaptive constraint modification. It uses probability estimation in the latent space to select instance-specific constraints for modification, preserving core characteristics. It also groups strongly related constraints via community detection for collective modification, aiming to maintain constraint inter-relations and improve hardness preservation.

**Diffusion-based Approaches:** Recognizing the power of diffusion models in generation tasks, MILP-FBGen (Zhang et al., 2024) proposed a diffusion-based framework. It uses a structure-preserving module to modify constraint/variable vectors and a feasibility/boundedness-constrained sampling module for the right-hand side and objective coefficients, aiming to ensure both structural similarity and feasibility/boundedness. Another diffusion-based approach focuses specifically on generating *feasible solutions* (rather than full instances) using guided diffusion. It employs contrastive learning to embed instances and solutions and uses a diffusion model conditioned on instance embeddings to generate solution embeddings, guided by constraints and objectives during sampling.

**Structure-Focused Approaches:** Some recent works explicitly target specific structural properties. MILP-StuDio (Liu et al., 2024) focuses on preserving and manipulating *block structures* commonly found in CCMs of real-world MILPs. It decomposes instances into block units, builds a library of these units, and uses operators (reduction, mix-up, expansion) to construct new, potentially scalable instances while maintaining structural integrity and properties like hardness and feasibility. MILP-Gen (Yang et al., 2024b) also aims at scalability, using node splitting/merging on the bipartite graph representation to decompose instances into tree structures, which are then concatenated to build larger problems.

These learning-based methods offer the potential to generate diverse and realistic MILP instances automatically, addressing the data scarcity issue. The generated instances have shown promise in downstream tasks like solver hyperparameter tuning, data augmentation for ML models predicting optimal values or guiding solvers, and potentially constructing harder benchmarks.

## C  PARAMETER SUMMARY

See the parameters of our methods in Table 9.

Table 9: Summary of Key Experimental Parameters.

| Category | Parameter Name | Value | Description |
|---|---|---|---|
| **Feasibility Ratio** | Gurobi `time_limit` | 300 s | Maximum Gurobi solve time per instance for feasibility/boundedness check. |
| | Gurobi `threads` | 64 | Number of CPU threads used per Gurobi solve. |
| **Structural Similarity** | Feature Set | 11 features | Set of graph-based structural features extracted (e.g., density, degree stats). |
| | Comparison Metric | JSD-based | Jensen-Shannon Divergence $(1 - JSD/\log(2))$ averaged. |
| **Branching Nodes** | Gurobi `time_limit` | 300 s | Maximum Gurobi solve time per instance. |

Table 9: Summary of Key Experimental Parameters (Continued).

| Category | Parameter Name | Value | Description |
|---|---|---|---|
| | Gurobi `threads` | 1 | Threads per solve (to enhance determinism). |
| | Metric | Relative Error | $\lvert(\sum N_{gen} - \sum N_{train})/\sum N_{train}\rvert \times 100\%$. |
| **Solving Time Gap** | Gurobi `time_limit` | 3600 s | Maximum Gurobi solve time per instance. |
| | Gurobi `threads` | 256 | Threads per solve (high count to expedite experiments). |
| | Metric | Relative Difference | $\lvert(\text{mean\_}t_{gen} - \text{mean\_}t_{train})/\text{mean\_}t_{train}\rvert \times 100\%$. |
| **Hyperparameter Tuning** | Tuning Framework | SMAC3 | Automated algorithm configuration tool used. |
| | `n_trials` | 50 | SMAC3 evaluation budget. |
| | Objective | Mean Solve Time | Minimize mean wall-clock time on the tuning set. |
| | Gurobi `time_limit` | 60 s | Gurobi time limit per solve during tuning/evaluation. |
| | Gurobi `threads` | 256 | Gurobi threads per solve during tuning/evaluation. |
| | Tuned Parameters | Gurobi Controls | 8 parameters including `Heuristics`, `MIPFocus`, etc. |
| **Initial Basis Prediction** | *GNN Model Parameters* | | |
| | `num_layers` | 3 | Number of GNN message passing layers. |
| | `hidden_dim` | 128 | Dimensionality of hidden layers in the GNN. |
| | `dropout` | 0.1 | Dropout rate used during training. |
| | *GNN Training Parameters* | | |
| | `epochs` | 800 (Max) | Maximum number of training epochs. |
| | `batch_size` | 32 | Number of instances per training batch. |
| | `learning_rate` | 1e-3 | Initial learning rate for the Adam optimizer. |
| | `weight_decay` | 1e-4 | L2 regularization coefficient. |
| | `early_stopping` | 50 | Patience (epochs) for early stopping. |
| | Loss Function | Weighted CE | Weighted Cross-Entropy + Label Smoothing (0.1). |
| | *Evaluation Parameters* | | |
| | Gurobi `time_limit` | 600 s | Gurobi time limit per solve during evaluation. |
| | Gurobi `threads` | 256 | Gurobi threads per solve during evaluation. |
| | Basis Repair Threshold | 1e-12 | Pivot threshold for basis repair stability check. |
| **Solver-Internal Features** | *Data Generation Phase* | | |
| | Gurobi `time_limit` | 100 s | Gurobi time limit for log generation. |
| | Gurobi `threads` | 128 | Gurobi threads for log generation. |

Table 9: Summary of Key Experimental Parameters (Continued).

| Category | Parameter Name | Value | Description |
|---|---|---|---|
| | *Comparison Phase* | | |
| | Root Gap Metric | 1-Wasserstein | Metric to compare scalar distributions. |
| | Cut Plane Metric | PCA + 1-W. | Method for comparing high-dim cut vectors. |
| | PCA Components | 3 | Number of components retained for cut analysis. |
| | PCA Scaling | Standard | Data scaling method (Z-score) before PCA. |

# D   EXPERIMENT ENVIRONMENT

All experiments were conducted on a Linux-based system equipped with an AMD EPYC 9754 128-Core Processor with a typical operating frequency around 3.1 GHz and supports 256 threads (128 cores with 2 threads per core). The system has 256 MiB of L3 cache and 251 GiB of system RAM.

For GPU-accelerated computations, a NVIDIA GeForce RTX 3090 graphics card with 24 GiB of VRAM was utilized. The NVIDIA driver version was 535.171.04, supporting CUDA up to version 12.2. The CUDA Toolkit version used for development and compilation was 11.8 (V11.8.89).

The operating system was Ubuntu 20.04.6 LTS. Key software includes Python 3.8.20, Gurobi Optimizer version 11.0.1.

# E   MORE PRELIMINARIES

## E.1   FUNDAMENTAL OPTIMIZATION CONCEPTS

**Feasibility**   Consider the MILP problem defined by constraints $\mathbf{A}\mathbf{x} \leq \mathbf{b}$ and $x_i \in \mathbb{Z}, \forall i \in \mathcal{I}$. The feasible region is the set $\mathcal{F} = \{\mathbf{x} \in \mathbb{R}^n \mid \mathbf{A}\mathbf{x} \leq \mathbf{b}, x_i \in \mathbb{Z} \, \forall i \in \mathcal{I}\}$. The MILP problem is *feasible* if its feasible region $\mathcal{F}$ is non-empty, i.e., $\mathcal{F} \neq \emptyset$. A specific point $\hat{\mathbf{x}} \in \mathbb{R}^n$ is considered *feasible* if it satisfies all constraints, i.e., $\hat{\mathbf{x}} \in \mathcal{F}$.

**Boundedness**   Given the MILP objective $\min_{\mathbf{x}\in\mathcal{F}} \tilde{\mathbf{c}}^\top \mathbf{x}$, the problem is *bounded* if the optimal objective value $z^*$ is finite. This means $z^* = \inf_{\mathbf{x}\in\mathcal{F}} \tilde{\mathbf{c}}^\top \mathbf{x} > -\infty$. (For a maximization problem $\max_{\mathbf{x}\in\mathcal{F}} \tilde{\mathbf{c}}^\top \mathbf{x}$, boundedness would mean $z^* = \sup_{\mathbf{x}\in\mathcal{F}} \tilde{\mathbf{c}}^\top \mathbf{x} < +\infty$). If a problem is both feasible ($\mathcal{F} \neq \emptyset$) and bounded, a finite optimal objective value $z_{MILP}$ exists.

## E.2   CORE SOLUTION ALGORITHMS AND TECHNIQUES

**Simplex Method Basics**   The **Simplex Method** (Dantzig, 1963) is an algorithm primarily designed for solving Linear Programming (LP) problems, such as the LP relaxation defined above. While it can handle various forms, it often operates by converting the problem to a standard form like $\min\{\mathbf{c'}^\top \mathbf{y} \mid \mathbf{A'}\mathbf{y} = \mathbf{b'}, \mathbf{y} \geq 0\}$ (where $\mathbf{y}$ might include original and slack/surplus variables). It explores the vertices (Basic Feasible Solutions, BFS) of the feasible polyhedron $\mathcal{P'} = \{\mathbf{y} \mid \mathbf{A'}\mathbf{y} = \mathbf{b'}, \mathbf{y} \geq 0\}$. It iteratively moves between adjacent BFS by exchanging one variable in the current basis (a set of variables determining the BFS) with a non-basic variable, guided by criteria like reduced costs to ensure monotonic improvement of the objective function value, until an optimal BFS is found or unboundedness is detected.

**Heuristics**   Within MILP solvers, **heuristics** (Berthold, 2012) are algorithms designed to rapidly find a feasible integer solution $\hat{\mathbf{x}} \in \mathcal{F}_{MILP} = \{\mathbf{x} \mid \mathbf{A}\mathbf{x} \leq \mathbf{b}, x_i \in \mathbb{Z} \, \forall i \in \mathcal{I}\}$, whose objective value $\tilde{\mathbf{c}}^\top \hat{\mathbf{x}}$ is hopefully close to the true optimum $z_{MILP}$. They do not guarantee optimality. Their main purpose is to quickly establish strong primal bounds (updating the incumbent $z_{incumbent} = \min(z_{incumbent}, \tilde{\mathbf{c}}^\top \hat{\mathbf{x}})$). In branch-and-bound, these incumbents enable pruning of search nodes $j$ where the local lower bound $z_{bound}^{(j)}$ satisfies $z_{bound}^{(j)} \geq z_{incumbent}$.

**Cutting Planes (Cuts)**       **Cutting planes** (Conforti et al., 2014) are linear inequalities, $\alpha^T \mathbf{x} \leq \beta$, that are valid for all feasible integer solutions, i.e., they hold for all $\mathbf{x} \in Conv(\mathcal{F}_{MILP})$ where $\mathcal{F}_{MILP} = \{\mathbf{x} \mid \mathbf{A}\mathbf{x} \leq \mathbf{b}, x_i \in \mathbb{Z} \ \forall i \in \mathcal{I}\}$. However, they are chosen such that they are violated by the optimal solution $\mathbf{x}^*_{LP}$ of the current LP relaxation (i.e., $\alpha^T \mathbf{x}^*_{LP} > \beta$). Adding such cuts $\mathcal{C}$ to the LP relaxation feasible set $\mathcal{P} = \{\mathbf{x} \mid \mathbf{A}\mathbf{x} \leq \mathbf{b}\}$ yields a tighter relaxation $\mathcal{P}' = \mathcal{P} \cap \{\mathbf{x} \mid \alpha^T \mathbf{x} \leq \beta$ for all, $\alpha^T \mathbf{x} \leq \beta \in \mathcal{C}\}$. The goal is to improve the lower bound derived from the relaxation, $\min_{\mathbf{x} \in \mathcal{P}'} \tilde{\mathbf{c}}^\top \mathbf{x} \geq z_{LP}$, thereby bringing it closer to the true MILP optimum $z_{MILP}$ and potentially reducing the search space.

## F  FEASIBILITY

Below is the result of feasibility test.

Table 10: Feasibility and Boundedness Ratios Across All Problem Types. Each configuration was tested on 1000 instances.

| Problem | Model | Ratio ($\eta$) | Feasible (%) |
|---|---|---|---|
| Independent Set | Original Set | N/A | 100.00 |
| | G2MILP | 0.01 | 100.00 |
| | | 0.05 | 100.00 |
| | | 0.10 | 93.40 |
| | ACM-MILP | 0.05 | 100.00 |
| | | 0.10 | 100.00 |
| | | 0.20 | 100.00 |
| Combinatorial Auction | Original Set | N/A | 100.00 |
| | ACM-MILP | 0.05 | 100.00 |
| | | 0.10 | 100.00 |
| | | 0.20 | 100.00 |
| | DIG-MILP | 0.01 | 100.00 |
| | | 0.05 | 100.00 |
| | | 0.10 | 100.00 |
| Set Cover | Original Set | N/A | 100.00 |
| | G2MILP | 0.01 | 100.00 |
| | | 0.05 | 100.00 |
| | | 0.10 | 100.00 |

## G  BRANCHING NODES

Table 11 are the detailed results of branching nodes, which is not revealed in main tex.

## H  STRUCTURAL SIMILARITY

The experiment's objective is to quantitatively compare the structural similarity between two sets of MILP instances. This is achieved by extracting graph-based structural features from each instance and then calculating the Jensen-Shannon Divergence (JSD) to measure the difference between the feature distributions of the two sets.

### H.1  METHODOLOGY

**1. Feature Extraction:** Each MILP instance is converted into a bipartite graph. From this graph, a vector of structural features is extracted, capturing characteristics related to matrix sparsity, variable and constraint connectivity, coefficient statistics, and other graph-theoretic properties. This process is computationally parallelized for efficiency. The specific features are grouped into two categories and detailed in Table 12.

Table 11: Branching Node Statistics and Relative Error across Problem Types

| Problem | Model | Mask Ratio ($\eta$) | Mean Nodes | Median Nodes | Std Dev | Max Nodes | Time Limit Hits | Relative Error (%) |
|---|---|---|---|---|---|---|---|---|
| IS | Baseline (IS) | N/A | 14.3 | 1.0 | 49.1 | 1,056 | 0 | — |
| | G2MILP | 0.01 | 26.2 | 1.0 | 60.1 | 724 | 0 | 84.1 |
| | G2MILP | 0.05 | 765.7 | 175.0 | 1,941.6 | 16,314 | 5 | 5,271.6 |
| | G2MILP | 0.10 | 8,386.4 | 8,330.0 | 1,219.5 | 11,789 | 982 | 58,730.0 |
| | ACM-MILP | 0.05 | 8.8 | 1.0 | 36.8 | 572 | 0 | 38.3 |
| | ACM-MILP | 0.10 | 10.9 | 1.0 | 63.8 | 1,344 | 0 | 23.4 |
| | ACM-MILP | 0.20 | 12.5 | 1.0 | 78.0 | 1,977 | 0 | 12.3 |
| CA | Baseline (CA) | N/A | 18.5 | 1.0 | 48.1 | 514 | 0 | — |
| | ACM-MILP | 0.05 | 6,556.0 | 1,661.5 | 9,926.3 | 51,913 | 138 | 35,360.0 |
| | ACM-MILP | 0.10 | 6,504.9 | 1,547.0 | 9,962.8 | 54,385 | 120 | 35,084.3 |
| | ACM-MILP | 0.20 | 4,873.6 | 962.5 | 8,021.7 | 43,036 | 75 | 26,259.8 |
| | DIG-MILP | 0.01 | 34.1 | 1.0 | 76.2 | 799 | 0 | 84.4 |
| | DIG-MILP | 0.05 | 34.3 | 1.0 | 76.4 | 812 | 0 | 85.6 |
| | DIG-MILP | 0.10 | 37.8 | 1.0 | 83.5 | 902 | 0 | 104.3 |
| SC | Baseline (SC) | N/A | 46.4 | 1.0 | 298.6 | 8,135 | 0 | — |
| | G2MILP | 0.01 | 85.4 | 1.0 | 536.5 | 14,592 | 0 | 84.0 |
| | G2MILP | 0.05 | 79.6 | 1.0 | 430.9 | 10,643 | 0 | 71.6 |
| | G2MILP | 0.10 | 63.7 | 1.0 | 269.8 | 4,713 | 0 | 37.3 |

Note: Statistics calculated over 1000 instances per model/ratio. Baselines refer to original training sets (Total Nodes: IS=14,255; CA=18,488; SC=46,408). 'Time Limit Hits' indicates premature termination. Relative Error (RE) compares generated total nodes to baseline total nodes. $\eta$ denotes mask ratio.

Table 12: Structural Features Used for Similarity Analysis

| Category | Feature | Metric(s) | Description |
|---|---|---|---|
| Basic Graph Features | Coefficient Density | `coef_dens` | Reflects the sparsity of the constraint matrix. |
| | Variable Node Degree | `var_degree_mean/std` | Describes the occurrence patterns of variables in constraints. |
| | Constraint Node Degree | `cons_degree_mean/std` | Reflects the number of variables involved in each constraint. |
| | Coefficient Statistics | `lhs/rhs_mean/std` | Captures the numerical distribution of constraint terms. |
| Advanced Topological | Clustering Coefficient | `clustering` | Measures the local connection density between variables. |
| | Modularity | `modularity` | Assesses the degree of modularity in the problem structure. |

Table 13: **Solving Time Gap Comparison**. The table shows the percentage difference in average solving time between instances generated by different models (with varying mask ratios $\eta$) and the original training set instances.

| Training Set (Original Avg Time) | Model | $\eta$ | Avg Time (s) | Solving Time Gap (%) |
|---|---|---|---|---|
| *Set Cover (0.2644s)* | G2MILP | 0.10 | 0.3002 | **13.54** |
| | | 0.05 | 0.2945 | **11.38** |
| | | 0.01 | 0.3237 | 22.43 |
| *Combinatorial Auction (0.1020s)* | ACM-MILP | 0.20 | 2.6449 | 2493.04 |
| | | 0.10 | 3.1654 | 3003.33 |
| | | 0.05 | 3.4461 | **3278.53** |
| | DIG-MILP | 0.10 | 0.1312 | 28.63 |
| | | 0.05 | 0.1225 | **20.10** |
| | | 0.01 | 0.1237 | 21.27 |
| *Independent Set (0.3374s)* | ACM-MILP | 0.20 | 0.1722 | 48.96 |
| | | 0.10 | 0.1774 | 47.42 |
| | | 0.05 | 0.1788 | 47.01 |

**2. Per-Feature Similarity:** For each feature, the Jensen-Shannon Divergence is used to compare its distribution across the two sets of instances. This divergence value is then normalized to a similarity score ranging from 0 (maximal divergence) to 1 (identical distributions).

**3. Overall Similarity Score:** A final, overall similarity score is calculated by taking the arithmetic mean of all the individual feature similarity scores. This single metric represents the aggregate structural similarity between the two instance sets.

# I  SOLVING TIME GAP

This experiment assesses the similarity in computational hardness between a 'training' set and a 'generated' set of MILP instances. The metric is the relative percentage difference in their average solve times using the Gurobi optimizer.

**Methodology:** Each instance from both sets is solved under identical Gurobi configurations to ensure a fair comparison. The key parameters are:

- **Time Limit:** 300 seconds per instance.
- **Threads:** 256 per instance.

The average solve time is computed for each set. The final metric is calculated using the formula:

$$\text{Relative Time Gap} = \frac{|\text{generated\_mean\_time} - \text{training\_mean\_time}|}{\max(\text{training\_mean\_time}, 10^{-10})} \times 100\%$$

A lower percentage signifies a greater similarity in computational hardness between the two instance sets.

## I.1  RESULTS

The results are in Table 13. We have omitted the G2MILP-generated Independent Set (IS) instances from this presentation. The rationale for this decision lies in the substantial surge in their computational complexity compared to the baseline instances, leading to a prohibitive increase in solution times by a factor of several thousands.

# J  HYPERPARAMETER TUNING FOR GUROBI

The objective of this project is to automatically tune a key set of Gurobi solver hyperparameters using the SMAC3 framework. The goal is to find a parameter configuration that **minimizes the average wall-clock solve time** for a specific collection of MILP instances, designated as the **tuning set**. Subsequently, the performance of this optimized configuration is compared against Gurobi's default settings on a separate, unseen **test set** to evaluate its generalization capability.

## J.1 METHODOLOGY

The process is divided into five phases to ensure robust optimization and an unbiased evaluation.

**Phase 1: Inputs and Setup**  The process utilizes two independent sets of MILP instances: a **tuning set**, which is used to train the SMAC3 model and guide its search for optimal parameters, and a separate **test set**, which is held out exclusively for the final performance evaluation to ensure an objective assessment. The core tuning engine is **SMAC3**, which directly interfaces with the target solver, **Gurobi**.

**Phase 2: Hyperparameter Space Definition**  We selected eight critical Gurobi hyperparameters for optimization. The parameter space includes one continuous parameter, `Heuristics`, defined on the range [0.0, 1.0]. The remaining seven parameters are categorical: `MIPFocus` {0, 1, 2, 3}, `VarBranch` {-1, 0, 1, 2, 3}, `BranchDir` {-1, 0, 1}, `Presolve` {-1, 0, 1, 2}, `PrePasses` {-1, ..., 20}, `Cuts` {-1, 0, 1, 2, 3}, and `Method` {-1, ..., 5}.

**Phase 3: Automated Tuning with SMAC3**  SMAC3 performs an iterative optimization over a budget of **50 trials**. Each trial begins with SMAC3 suggesting a new hyperparameter configuration from its internal model. Gurobi then uses this configuration to solve all instances in the **tuning set**, with each solve constrained to a **60-second time limit** and using **256 threads**. The resulting **average solve time** across the set is calculated and returned to SMAC3 as a cost metric. This feedback allows SMAC3 to update its model and inform its choice for the next trial. Upon completion, SMAC3 reports the **best configuration** found, which is the one that yielded the lowest average solve time.

**Phase 4: Performance Evaluation on the Test Set**  To assess generalization, we compare the **best configuration** found by SMAC3 against Gurobi's **default configuration** on the independent **test set**. Both configurations are run under identical computational conditions (60-second time limit, 256 threads) to ensure a fair and direct comparison of their performance.

**Phase 5: Outputs and Analysis**  The final outputs synthesize the results into two key components. The primary metric is a comparison of the average solve times on the test set for the best versus default configurations, often expressed as a percentage improvement. The analysis also provides the **best configuration** itself, detailing the specific values for the eight optimized hyperparameters that yielded the top performance.

Table 14 shows the best Gurobi hyperparameter values found by SMAC3 for each experiment. The default Gurobi parameter values are: `Heuristics`=0.05, `MIPFocus`=0, `VarBranch`=-1, `BranchDir`=0, `Presolve`=-1, `PrePasses`=-1, `Cuts`=-1, `Method`=-1.

Table 14: Best Gurobi hyperparameter values found for each experiment. Column Abbreviations: Heur.: Heuristics, Focus: MIPFocus, VarBr.: VarBranch, BrDir.: BranchDir, Pres.: Presolve, PreP.: PrePasses, Cuts: Cuts, Meth.: Method.

| Experiment Name | Heur. | Focus | VarBr. | BrDir. | Pres. | PreP. | Cuts | Meth. |
|---|---|---|---|---|---|---|---|---|
| acmmilp_mis_0.1 | 0.189 | 2 | 1 | -1 | -1 | 10 | 0 | 5 |
| acmmilp_mis_0.2 | 0.474 | 0 | 1 | 0 | -1 | 17 | 1 | 2 |
| acmmilp_mis_0.05 | 0.500 | 0 | -1 | -1 | -1 | -1 | -1 | -1 |
| ca | 0.186 | 1 | 1 | 1 | 0 | 1 | -1 | 4 |
| digmilp_ca_0.1 | 0.186 | 1 | 1 | 1 | 0 | 1 | -1 | 4 |
| digmilp_ca_0.05 | 0.186 | 1 | 1 | 1 | 0 | 1 | -1 | 4 |
| digmilp_ca_0.01 | 0.186 | 1 | 1 | 1 | 0 | 1 | -1 | 4 |
| mis | 0.498 | 0 | 1 | -1 | 1 | -1 | 1 | 3 |
| setcover | 0.189 | 2 | 1 | -1 | -1 | 10 | 0 | 5 |
| g2milp_setcover_0.01 | 0.189 | 2 | 1 | -1 | -1 | 10 | 0 | 5 |
| g2milp_setcover_0.05 | 0.189 | 2 | 1 | -1 | -1 | 10 | 0 | 5 |

# K    Initial Basis Prediction

Following the approach of (Fan et al., 2023), this experiment aims to leverage Graph Neural Networks (GNNs) to predict a high-quality initial basis for the LP relaxation of MILP instances. The primary goal is to train a GNN model, typically on a specified dataset (e.g., generated instances), and evaluate whether using the initial basis predicted by this trained model can accelerate the MILP solving process in Gurobi compared to Gurobi's default initialization. The evaluation measures performance improvements in terms of Gurobi runtime, node count, and iteration count on an unseen test set.

## K.1    Methodology & Operations

The initial basis prediction methodology employs a comprehensive six-phase approach that transforms MILP instances into graph representations, extracts meaningful features, trains sophisticated neural network models, and evaluates performance improvements. This systematic process ensures robust basis prediction capabilities while maintaining numerical stability throughout the optimization pipeline.

**Phase 1: Problem Description and Data Representation**    The foundation of the approach rests on transforming MILP instances into structured graph representations suitable for neural network processing. Each MILP instance is characterized by its constraint matrix $A \in \mathbb{R}^{m \times n}$, right-hand side vector $b \in \mathbb{R}^m$, objective coefficients $c \in \mathbb{R}^n$, and variable bounds $l^x, u^x \in \mathbb{R}^n$. These mathematical components collectively define the optimization landscape that the GNN must learn to navigate effectively.

The core prediction task focuses on determining an optimal initial basis for the LP relaxation, which requires identifying a set of $m$ basic variables and slack variables. This basis is formally defined by the sets $B_x \subset \{1..n\}$ and $B_s \subset \{1..m\}$, where $|B_x| + |B_s| = m$, such that the corresponding basis matrix maintains non-singularity for numerical stability.

To enable GNN processing, each MILP instance undergoes transformation into a bipartite graph $G = (V, E)$. The vertex set $V$ comprises $n$ variable nodes representing decision variables and $m$ constraint nodes representing problem constraints. The edge set $E$ captures the structural relationships within the constraint matrix, where an edge $(v_i, w_j) \in E$ exists if and only if $A_{ji} \neq 0$, with the corresponding edge weight set to the matrix element value $A_{ji}$.

**Phase 2: Feature Engineering**    The feature engineering process creates rich node representations that capture both local structural properties and global problem characteristics. This dual-level feature extraction ensures that the GNN receives comprehensive information about each node's role within the optimization problem.

Variable node features encompass eight distinct characteristics designed to capture the economic and structural significance of each decision variable. The objective coefficient $c_i$ provides direct insight into the variable's contribution to the optimization objective. Variable density, calculated as $\text{nnz}(A_{:i})/m$, quantifies how extensively the variable participates in problem constraints.

The feature set further includes similarity measures that relate each variable to constraint bounds. The similarity to slack lower bounds is computed as $\langle A_{:i}, l^s \rangle / (\|A_{:i}\|\|l^s\|)$, while similarity to slack upper bounds follows the analogous formula $\langle A_{:i}, u^s \rangle / (\|A_{:i}\|\|u^s\|)$. Variable bounds are encoded through both their numerical values and binary flags: the lower bound $l_i^x$ (set to 0 if infinite) paired with a flag (-1 for $-\infty$, 0 for finite), and the upper bound $u_i^x$ (set to 0 if infinite) paired with its flag (1 for $+\infty$, 0 for finite).

Constraint node features mirror this comprehensive approach with eight corresponding characteristics tailored to capture constraint properties. The similarity to objective coefficients, computed as $\langle A_{j:}, c \rangle / (\|A_{j:}\|\|c\|)$, measures how closely each constraint aligns with the optimization objective. Constraint density $\text{nnz}(A_{j:})/n$ quantifies the breadth of variable involvement in each constraint.

Additional constraint features include similarities to variable bounds: $\langle A_{j:}, l^x \rangle / (\|A_{j:}\|\|l^x\|)$ for lower bounds and $\langle A_{j:}, u^x \rangle / (\|A_{j:}\|\|u^x\|)$ for upper bounds. Constraint slack bounds follow the

same encoding pattern as variable bounds, with numerical values $l_j^s$ and $u_j^s$ (set to 0 if infinite) accompanied by their respective flags.

Feature normalization ensures numerical stability and training efficiency. All continuous features, excluding the binary flags, undergo z-score standardization before input to the neural network, creating zero-mean, unit-variance distributions that facilitate effective gradient-based learning.

**Phase 3: Graph Neural Network Model**   The neural architecture centers on an initial GNN model specifically designed to handle bipartite graph structures inherent in MILP representations. The architecture follows a hierarchical design that progressively refines node embeddings through multiple message-passing layers.

The model begins with initial MLP layers that project the 8-dimensional input features into a 128-dimensional hidden space. This projection creates a rich embedding space where structural and semantic relationships can be effectively captured and manipulated.

The core computational component consists of $L = 3$ `BipartiteMessagePassing` layers that implement iterative message passing between variable and constraint nodes. Each layer updates node embeddings by aggregating information from neighboring nodes, enabling the model to capture both local structural patterns and global problem characteristics. Residual connections are incorporated throughout these layers to facilitate gradient flow and enable deeper architectures.

The output stage employs separate MLP heads for variables and constraints, each producing 3-dimensional logits corresponding to the three possible basis states: NonbasicAtLower, Basic, and NonbasicAtUpper. This separation allows the model to learn distinct patterns for variable and constraint basis predictions.

Knowledge masking represents a crucial component that enforces physical constraints on the predictions. These masks prevent the model from predicting impossible basis states based on variable and constraint bounds, ensuring that all predictions remain feasible within the problem's mathematical constraints.

The final prediction probabilities emerge through softmax normalization applied to the masked logits, yielding probability distributions $p_{x_i}$ and $p_{s_j}$ over the three basis states for each variable and slack variable respectively. Dropout regularization (0.1) is applied throughout the architecture to prevent overfitting and improve generalization.

**Phase 4: Model Training**   The training process employs a comprehensive framework designed to learn effective basis prediction patterns from the training dataset. A single GNN model undergoes training using instances from the specified training directory, creating a unified predictor capable of handling diverse problem structures within the domain.

The `GNNTrainer` orchestrates the training process over a maximum of 800 epochs, processing data in batches of size 32. This extended training horizon allows the model to thoroughly explore the parameter space while the moderate batch size balances computational efficiency with gradient estimate quality.

The loss function employs a sophisticated design that addresses multiple training challenges simultaneously. Label smoothing (0.1) improves generalization by preventing overconfident predictions and encouraging the model to maintain uncertainty about borderline cases. Class weights, computed dynamically per batch, mitigate the inherent class imbalance in basis prediction tasks where basic variables typically represent a small fraction of all variables.

Optimization proceeds through the Adam optimizer with an initial learning rate of 1e-3 and L2 regularization via weight decay (1e-4). This configuration balances rapid initial learning with long-term stability, while the regularization prevents overfitting to training data patterns.

Early stopping monitoring provides an essential safeguard against overfitting by terminating training when validation loss fails to improve for 50 consecutive epochs. This mechanism ensures that the final model represents the optimal balance between training performance and generalization capability. The model weights achieving the best validation performance are preserved for subsequent evaluation and deployment.

**Phase 5: Initial Basis Generation and Repair**  The basis generation process transforms GNN predictions into numerically stable basis configurations suitable for solver initialization. This critical phase bridges the gap between machine learning predictions and the strict mathematical requirements of linear programming solvers.

Candidate selection follows a probability-based approach where the $m$ variables and slack variables with the highest predicted probability of being Basic are initially selected for the basis. This selection strategy leverages the model's confidence estimates to prioritize the most likely basic variables.

However, the selected candidates may not form a numerically stable basis matrix, necessitating a sophisticated repair process. The basis repair procedure employs an iterative approach that performs LU decomposition on the candidate basis matrix and systematically replaces columns that contribute to numerical instability. This process continues until a well-conditioned basis matrix emerges, ensuring that the solver initialization will not encounter numerical difficulties.

Nonbasic state encoding completes the basis specification by determining appropriate bound settings for all nonbasic variables and slack variables. The GNN's predictions guide this assignment, with variables set to NonbasicAtLower or NonbasicAtUpper based on their predicted probability distributions. This comprehensive state specification provides the solver with a complete starting point that respects both the problem's mathematical constraints and the learned patterns from the training data.

**Phase 6: Performance Evaluation**  The evaluation framework provides rigorous assessment of the GNN-predicted basis performance through controlled comparison with Gurobi's default initialization. This evaluation employs an independent test set to ensure unbiased performance assessment and robust generalization estimates.

The evaluation protocol follows a systematic approach where each test instance undergoes solving under two different initialization strategies. The baseline approach utilizes Gurobi's default initialization procedures, representing the current state-of-the-art in commercial solver technology. The predicted approach employs the repaired basis generated by the trained GNN model.

Performance assessment encompasses multiple metrics that capture different aspects of solving efficiency. Runtime measurements provide the most direct indication of practical improvement, while node count and iteration count offer insights into the algorithmic efficiency gains achieved through superior initialization. These complementary metrics enable comprehensive understanding of where and why the GNN approach provides benefits.

Table 15: GNN prediction initial basis: Runtime and performance improvement

| Dataset | Source/Model | $\eta$ | Default Time (s) | Best Time (s) | Improvement (%) |
|---|---|---|---|---|---|
| CA | ACM-MILP | 0.05 | 0.078 | 0.077 | 2.6% |
| | | 0.10 | 0.079 | 0.076 | 2.6% |
| | | 0.20 | 0.078 | 0.075 | 4.0% |
| | DIG-MILP | 0.05 | 0.079 | 0.076 | 3.5% |
| | | 0.10 | 0.080 | 0.076 | 4.1% |
| | Original Dataset | — | 0.078 | 0.077 | 2.5% |
| IS | ACM-MILP | 0.10 | 0.236 | 0.239 | -0.2% |
| | G2MILP | 0.05 | 0.237 | 0.240 | -0.6% |
| | Original Dataset | — | 0.236 | 0.237 | 0.2% |
| SC | G2MILP | 0.01 | 0.236 | 0.221 | 10.6% |
| | | 0.05 | 0.240 | 0.229 | 9.1% |
| | | 0.10 | 0.242 | 0.224 | 12.4% |
| | Original Dataset | — | 0.236 | 0.226 | 9.1% |

## L  SOLVER-INTERNAL FEATURES

This experiment analyzes the behavioral patterns of the Gurobi solver to compare original and generated sets of MILP instances. The methodology involves a two-phase process: first, we gather

detailed performance metrics by solving each instance, and second, we apply statistical techniques to quantify the similarity between the instance sets based on the solver's behavior.

METHODOLOGY

**Phase 1: Data Generation and Metric Extraction**  In the first phase, we solve every MILP instance from both the original and generated collections using Gurobi under identical, standardized settings. This controlled environment ensures that any observed differences in performance are attributable to the instances themselves, not the solver's configuration. We enable detailed logging to capture a comprehensive record of the solver's internal operations.

From these logs, we systematically parse and extract key operational metrics. These metrics include the **duality gap at the root node**, which indicates initial problem difficulty; the **success counts of various heuristics**, which measure the effectiveness of the solver's solution-finding strategies; and the **usage frequency of different cut plane types** (e.g., Gomory, Cover, MIR), which reveals the solver's approach to tightening the LP relaxation. This process transforms the raw log files into structured data, ready for quantitative analysis.

**Phase 2: Comparative Statistical Analysis**  The second phase quantifies the similarity between the original and generated instance sets by comparing the distributions of the metrics collected in Phase 1. We primarily use the **1-Wasserstein distance** as a robust measure of distributional difference—a smaller distance implies greater similarity.

For one-dimensional metrics like the root node gap and heuristic success counts, we directly compare their distributions. For the multi-dimensional cut plane data, we first normalize the usage counts for each instance into proportions. We then apply **Principal Component Analysis (PCA)** to reduce the dimensionality of this data, focusing the comparison on the most significant patterns of solver behavior. The Wasserstein distance is then used to compare the instance sets along these principal components. The final output consists of these distance values, which serve as a quantitative score of behavioral similarity between the two datasets.

Table 16: Root Node Gap Statistics Comparison (Dataset Halves)

| Statistic | part1 | part2 |
|---|---|---|
| count | 500 | 500 |
| mean | 3.4637 | 3.4194 |
| std | 1.4386 | 1.4514 |
| min | 0.9722 | 1.0417 |
| 25% | 2.2222 | 2.4094 |
| 50% | 3.4483 | 3.2076 |
| 75% | 4.6289 | 4.5045 |
| max | 6.1364 | 6.5909 |
| W-Dist. | 0.1884 | 0.1884 |

Table 17: Heuristic Success Count Statistics Comparison (Dataset Halves)

| Statistic | part1 | part2 |
|---|---|---|
| count | 500 | 500 |
| mean | 6.0380 | 6.0400 |
| std | 1.7879 | 1.9541 |
| min | 2.0000 | 2.0000 |
| 25% | 5.0000 | 4.0000 |
| 50% | 6.0000 | 6.0000 |
| 75% | 7.0000 | 7.0000 |
| max | 11.0000 | 11.0000 |
| W-Dist. | 0.1300 | 0.1300 |

Table 18: Cut Plane Usage PCA 1-Wasserstein Distances (Dataset Halves)

| Principal_Component | W-Dist. |
|---|---|
| PC1 | 0.2318 |
| PC2 | 0.3295 |
| PC3 | 0.2280 |

Table 19: Overall Gurobi Solver Metrics Across Datasets

| Dataset | Avg. Root Gap (%) | Heuristic Successes |
|---|---|---|
| IS (Raw) | 3.42 | 18883 |
| Combinatorial Auction (Raw) | 2.47 | 2115 |
| Set Cover (Raw) | 7.61 | 2702 |
| ACM-MILP CA ($\eta = 0.1$) | 7.49 | 2552 |
| ACM-MILP IS ($\eta = 0.1$) | 2.66 | 1746 |
| DIG-MILP CA ($\eta = 0.05$) | 1.09 | 2241 |
| G2MILP SetCover ($\eta = 0.05$) | 7.75 | 2600 |

## L.1 VISUALIZATION

Here are some figures 3 4,5,6 showing the results of solver internal features comparison.

## L.2 EFFICIENCY TEST ON SUPER HARD INSTANCES

### L.2.1 EXPERIMENTAL OBJECTIVE

To validate the practical feasibility and robustness of our solver-internal feature analysis method, we designed an efficiency test. The core goal of this experiment is to demonstrate that even under strict computational constraints (i.e., tight time limits) and when facing a well-known set of hard instances, our method can still stably and efficiently extract meaningful feature distributions. This supports our key hypothesis that obtaining such deep behavioral features does not require solving instances to optimality, thereby making the evaluation both low-cost and highly efficient.

### L.2.2 EXPERIMENTAL METHODOLOGY

**Dataset Selection**: We used the `item_placement` dataset from the ML4CO competition, which is recognized for its high solving difficulty. A total of 1000 instances were selected for testing.

**Computational Budget Constraint**: To simulate scenarios with limited computational resources, we imposed a strict time limit of **120 seconds** for the Gurobi solver on each instance.

**Validation Method**: We employed a split-half validation approach to evaluate the stability of the extracted features:

• The 1000 solver logs were randomly divided into two halves, each containing 500 instances.

• We independently extracted solver-internal features from each half, including Root Node Gap, Heuristic Success Count, and Cut Plane Usage.

• We then computed and compared the distributional similarity of these features across the two halves. If the random subsets originating from the same underlying distribution exhibit highly similar feature distributions (i.e., low Wasserstein distance), it indicates strong stability and reliability of the feature-extraction method.

Table 20: Aggregated Cut Plane Counts (Sum over 1000 instances)

| Dataset | Gomory | ZeroHalf | Clique | MIR | RLT | FlowCover | Cover | ModK | RelaxLift | InfProof | StrongCG | ImplBound |
|---|---|---|---|---|---|---|---|---|---|---|---|---|
| IS (Raw) | 1305 | 61590 | 89 | 58 | 43763 | 0 | 0 | 0 | 0 | 0 | 0 | 0 |
| Combinatorial Auction (Raw) | 11183 | 7014 | 8941 | 98 | 81 | 0 | 3 | 24 | 0 | 0 | 6 | 0 |
| Set Cover (Raw) | 1350 | 3473 | 80 | 5601 | 240 | 0 | 0 | 8 | 0 | 0 | 0 | 1 |
| ACM-MILP CA ($\eta = 0.1$) | 6524 | 3132 | 26779 | 25 | 2 | 229 | 86 | 3 | 4 | 3 | 0 | 0 |
| ACM-MILP IS ($\eta = 0.1$) | 1087 | 34032 | 33 | 13 | 22932 | 0 | 0 | 1 | 0 | 0 | 0 | 0 |
| DIG-MILP CA ($\eta = 0.05$) | 6024 | 5629 | 462 | 531 | 114 | 0 | 14 | 34 | 0 | 1 | 14 | 0 |
| G2MILP SetCover ($\eta = 0.05$) | 1330 | 3228 | 31 | 5454 | 209 | 0 | 0 | 8 | 0 | 0 | 0 | 0 |

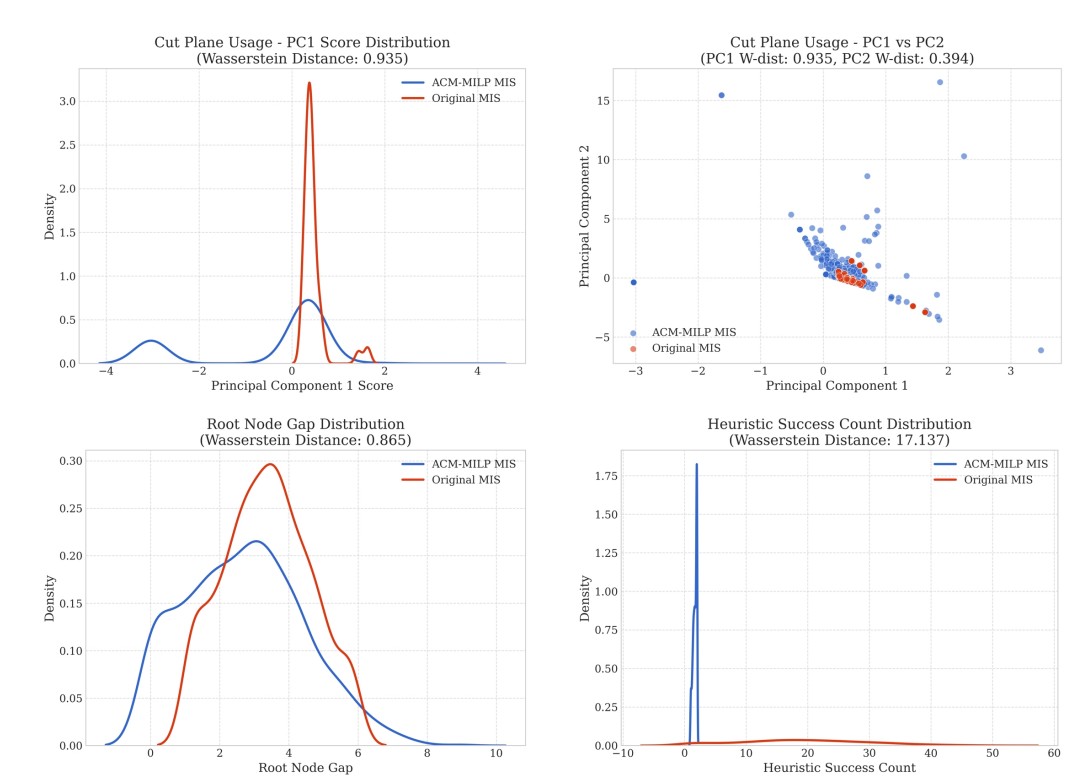

Figure 3: ACM-MILP IS at $\eta = 0.1$ compared with Original Dataset

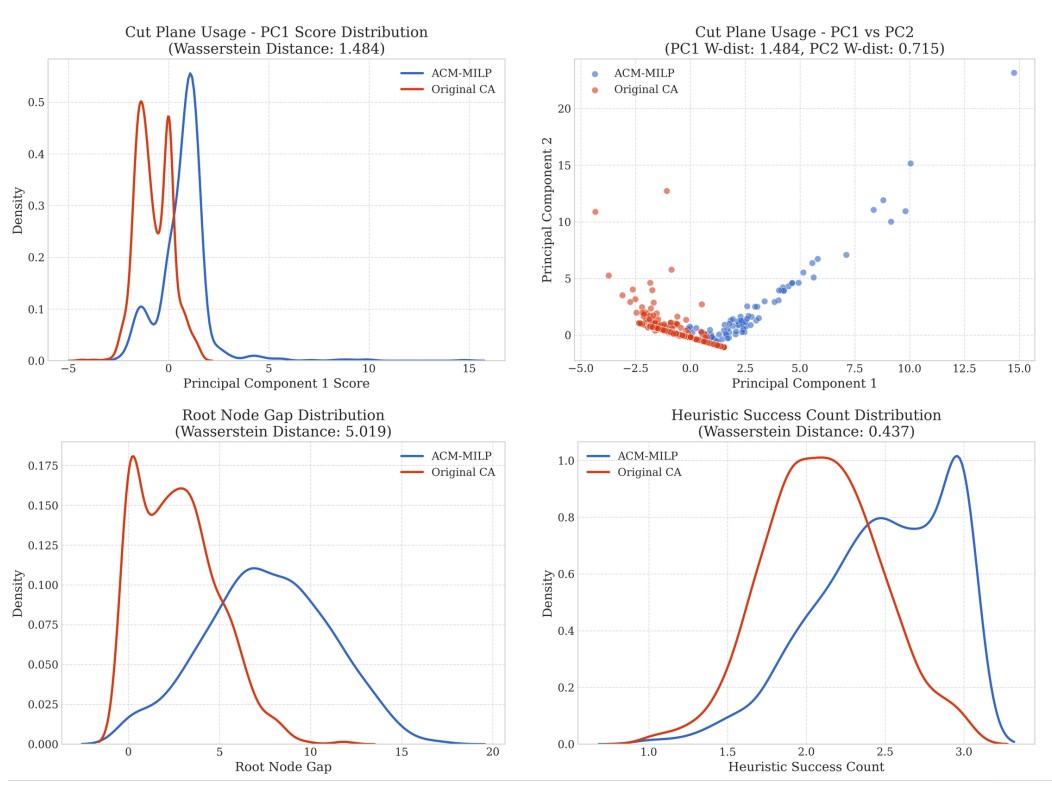

Figure 4: ACM-MILP CA at $\eta = 0.1$ compared with Original Dataset

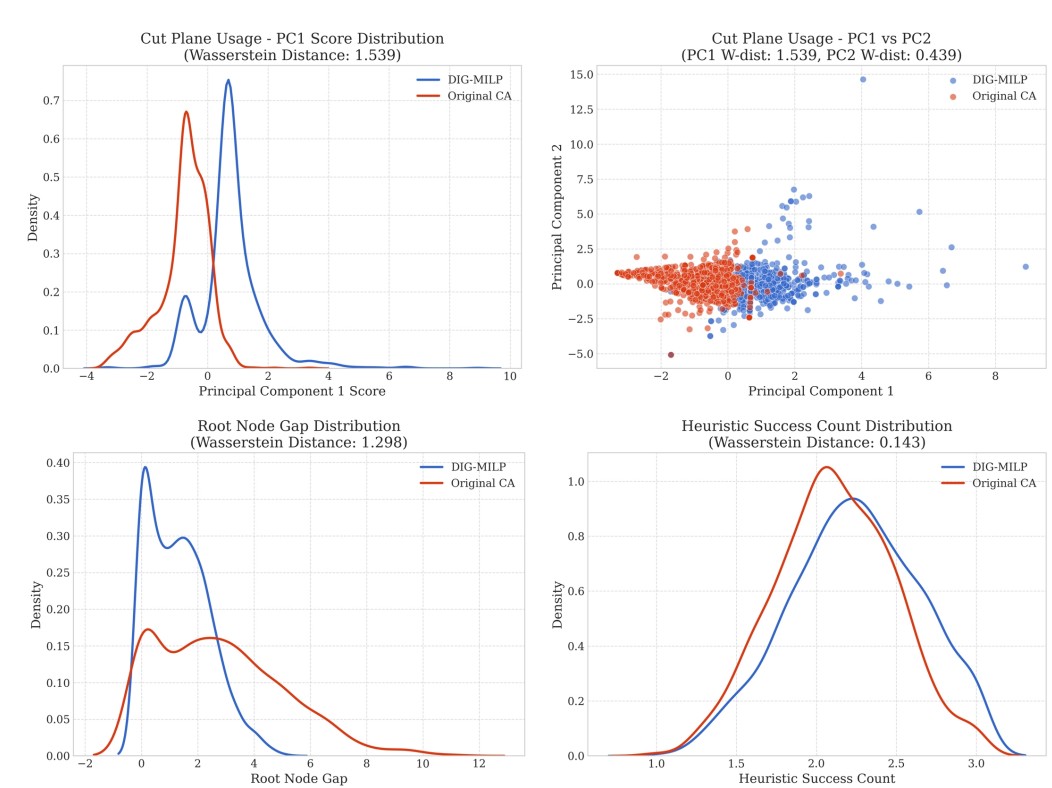

Figure 5: DIG-MILP CA at $\eta = 0.1$ compared with Original Dataset

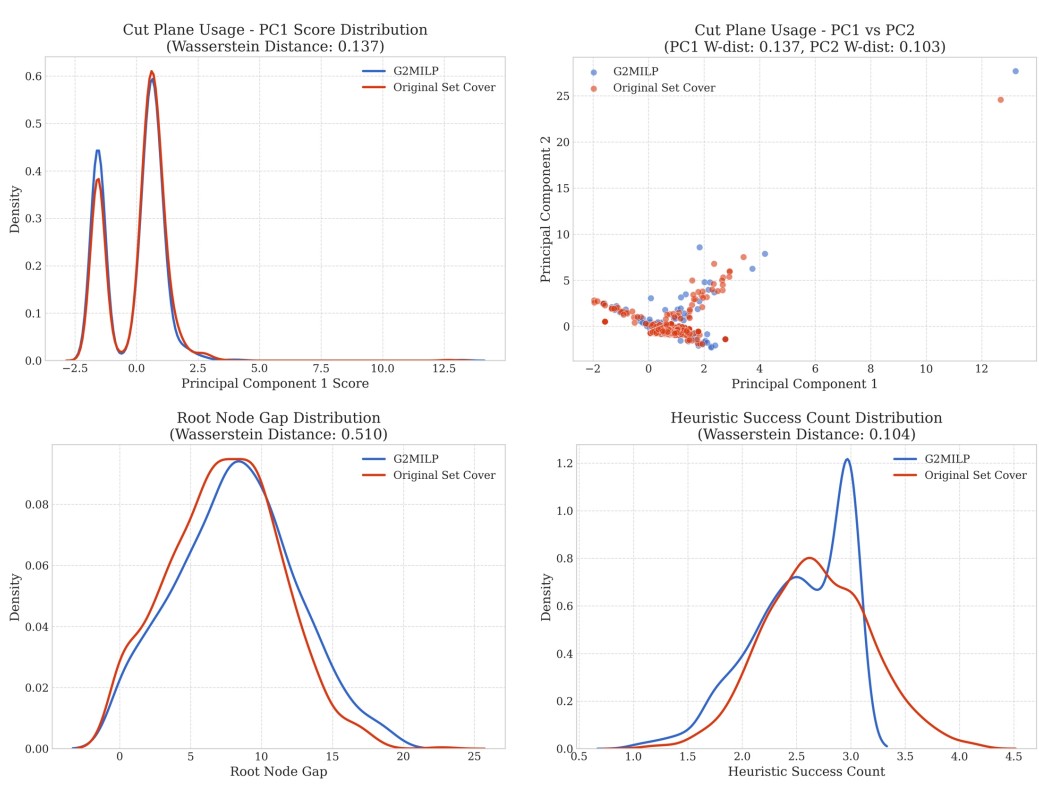

Figure 6: G2MILP SC at $\eta = 0.1$ compared with Original Dataset

Table 21: W-1 distances of key solver-internal features across two random halves

| Feature Dimension | W-1 Dist. | Remarks |
|---|---|---|
| Root Node Gap | 0.7613 | Mean difference only 0.04%; identical standard deviation |
| Heuristic Success Count | 0.7320 | Mean and standard deviation are very close |
| Cut Plane Usage | | Compared on top 3 principal components after PCA |
| – PC1 | 0.0917 | Extremely low distance indicates highly consistent cut-plane usage pattern |
| – PC2 | 0.0870 | |
| – PC3 | 0.0904 | |

### L.2.3 Experimental Results

The split-half experiment clearly shows that even under strict time constraints, the distributions of solver-internal features extracted from the solving process remain highly stable. Key indicators and their 1-Wasserstein distances are summarized in Table 21.

### L.2.4 Conclusion

This experiment strongly demonstrates the practical value of our framework:

- **High Efficiency and Low Cost**: The results confirm that a truncated solving process (120-second timeout) is sufficient to extract stable and representative solver-internal feature distributions. This substantially reduces the time and computational cost of evaluating large-scale or extremely hard instance sets.

- **Feature Stability**: The very small Wasserstein distances observed for Root Node Gap and Cut Plane Usage indicate that these early-stage solver behavior features are highly stable.

- **Practical Feasibility**: The success of this test shows that the EVA-MILP framework is not only theoretically sound but also practically applicable for evaluating MILP instances that are difficult to solve to optimality within a reasonable time frame and that reflect real-world complexity.

### L.3 Tests on Other Solvers

### L.3.1 Experimental Objective

To validate the generality of the EVA-MILP framework and to investigate potential differences in solver-internal behavioral features, we applied our core solver-internal feature analysis method to two leading open-source solvers: **SCIP** and **HiGHS**. This experiment is designed to answer two key questions:

1. *Can our framework be flexibly adapted to solvers other than Gurobi?*

2. *Do different solvers exhibit varying stability in their internal behaviors when solving the same instance set?*

### L.3.2 Experimental Methodology

We adopted the same split-half validation method used in the Gurobi experiments. Using the original Independent Set (IS) dataset, we randomly divided the instances into two subsets of 500 instances each. SCIP and HiGHS were then used to solve the two subsets separately, extracting three categories of solver-internal features: Root Node Gap, Heuristic Success, and Cut Plane Usage. Finally, we computed the 1-Wasserstein distance of these feature distributions between the two subsets to quantify the stability of each solver's internal behavior. For a stable solver, the Wasserstein distance between random subsets originating from the same distribution should remain very small.

### L.3.3 Experimental Results and Analysis

Our results show that, while the framework can be successfully applied to all tested solvers, their internal feature stability varies substantially.

Figure 7: Comparison of Root Node Gap distributions for SCIP in split-half validation. The two distributions represent the two random halves of the instance set.

**SCIP Solver**  Here are some figures 7, 8 and 9 showing the function of SCIP in spilt-half validation. For SCIP, the split-half experiment yields the following:

- **Root Node Gap**: The two subsets exhibit mean gaps of 7.13% and 8.88%, respectively. The Wasserstein distance is 1.753, which is much larger than the 0.1884 observed for Gurobi, indicating lower stability for SCIP in this metric.

- **Heuristic Success**: Principal Component Analysis (PCA) of six heuristic methods shows Wasserstein distances of PC1: 0.319, PC2: 0.426, and PC3: 0.778.

- **Cut Plane Usage**: PCA on four cut-plane types gives Wasserstein distances of PC1: 0.700, PC2: 0.298, and PC3: 0.241.

**HiGHS Solver**  Here are some figures 10, 11 and 12 showing the function of HiGHS in spilt-half validation. For HiGHS, the differences are even more pronounced:

- **Root Node Gap**: The two subsets have mean gaps of 364.28% and 386.68%, respectively. The Wasserstein distance reaches 26.7297, indicating extremely unstable root-node relaxation behavior.

- **Heuristic Success**: PCA analysis of heuristic methods gives Wasserstein distances of PC1: 0.2368 and PC2: 0.1695, with a weighted average of 0.1530, reflecting good stability.

- **Cut Plane Usage**: HiGHS shows very consistent cut-plane usage. PCA reveals that the first principal component explains 100% of the variance, with a Wasserstein distance of only 0.0612, indicating highly stable cut-plane strategies.

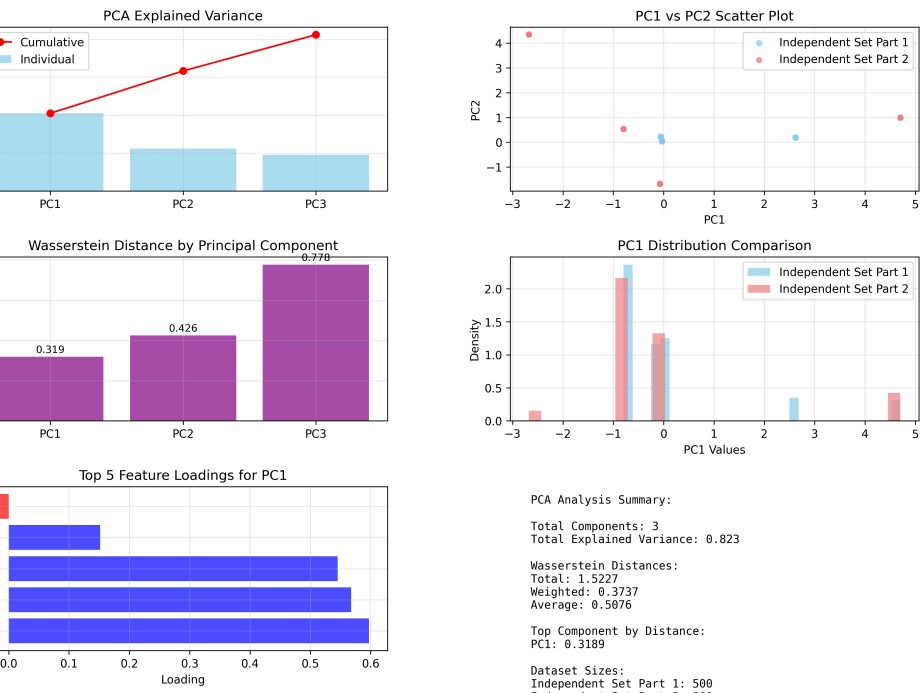

Figure 8: Comparison of Heuristic distributions for SCIP in split-half validation. The two distributions represent the two random halves of the instance set.

### L.3.4 CROSS-SOLVER COMPARISON

To facilitate comparison, Table 22 summarizes key 1-Wasserstein distances from the IS split-half validation for Gurobi, SCIP, and HiGHS.

Table 22: 1-Wasserstein distances across solvers on IS split-half validation

| Feature Dimension | Gurobi (Baseline) | SCIP | HiGHS |
|---|---|---|---|
| Root Node Gap | **0.1884** | 1.7530 | **26.7297** |
| Heuristic (PC1) | **0.1300** | 0.3190 | 0.2368 |
| Cut Plane (PC1) | 0.2318 | 0.7000 | **0.0612** |

From the table we draw three key conclusions:

- **Framework Generality**: EVA-MILP can be successfully applied to SCIP and HiGHS, confirming its potential as a general-purpose evaluation tool.

- **Stability Differences Across Solvers**: The most striking observation is the difference in Root Node Gap stability. Gurobi demonstrates exceptionally high stability, SCIP shows moderate instability, and HiGHS exhibits severe instability. This implies that solver strategies (e.g., preprocessing, initial LP relaxations) may have fundamentally different sensitivities to small instance perturbations.

- **Rationale for Choosing Gurobi as Baseline**: These findings strongly justify using Gurobi as the "expert evaluator" in our main study. Accurate and reliable instance-quality assessment demands a solver with highly stable internal behavior. Gurobi's stability enables precise detection of subtle computational similarities between instances, whereas instability in other solvers can obscure these signals.

In summary, although the EVA-MILP framework is solver-agnostic, the quality of its evaluation results is closely tied to the stability of the solver's internal strategies. State-of-the-art commercial solvers like Gurobi provide a more precise and reliable benchmark for our evaluation framework. At the same time, these experiments reveal an additional potential application of EVA-MILP: as

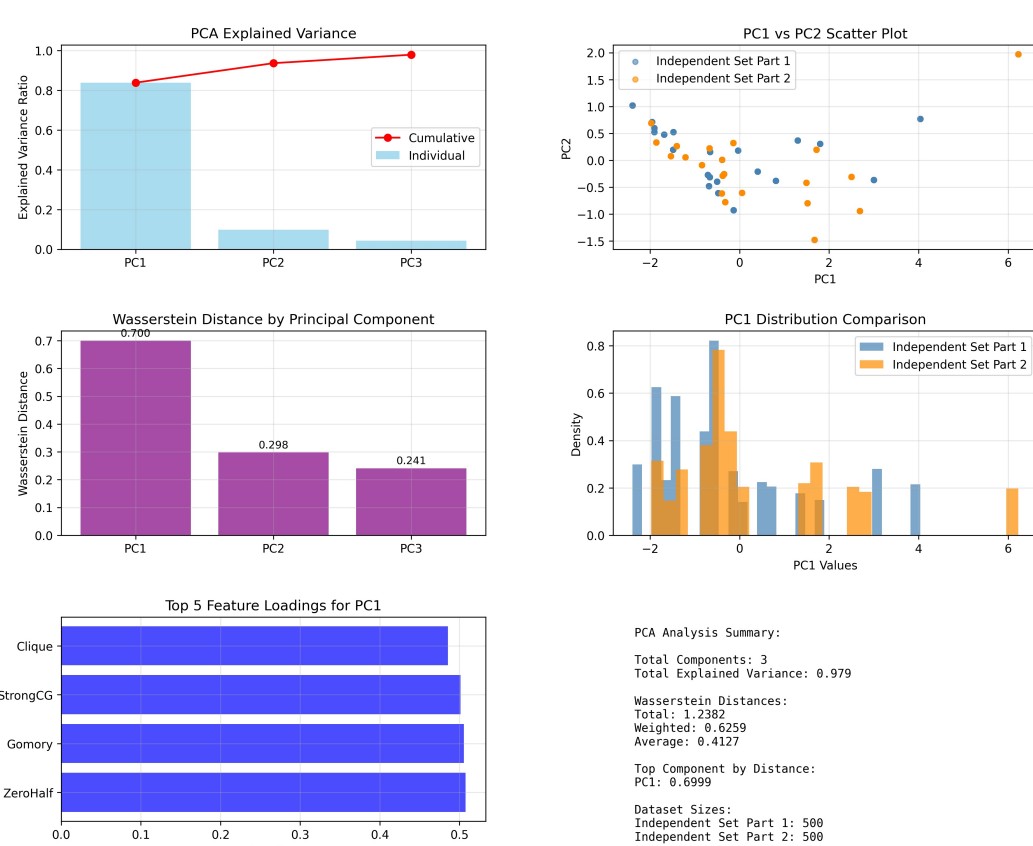

Figure 9: Comparison of cut plane distributions for SCIP in split-half validation. The two distributions represent the two random halves of the instance set.

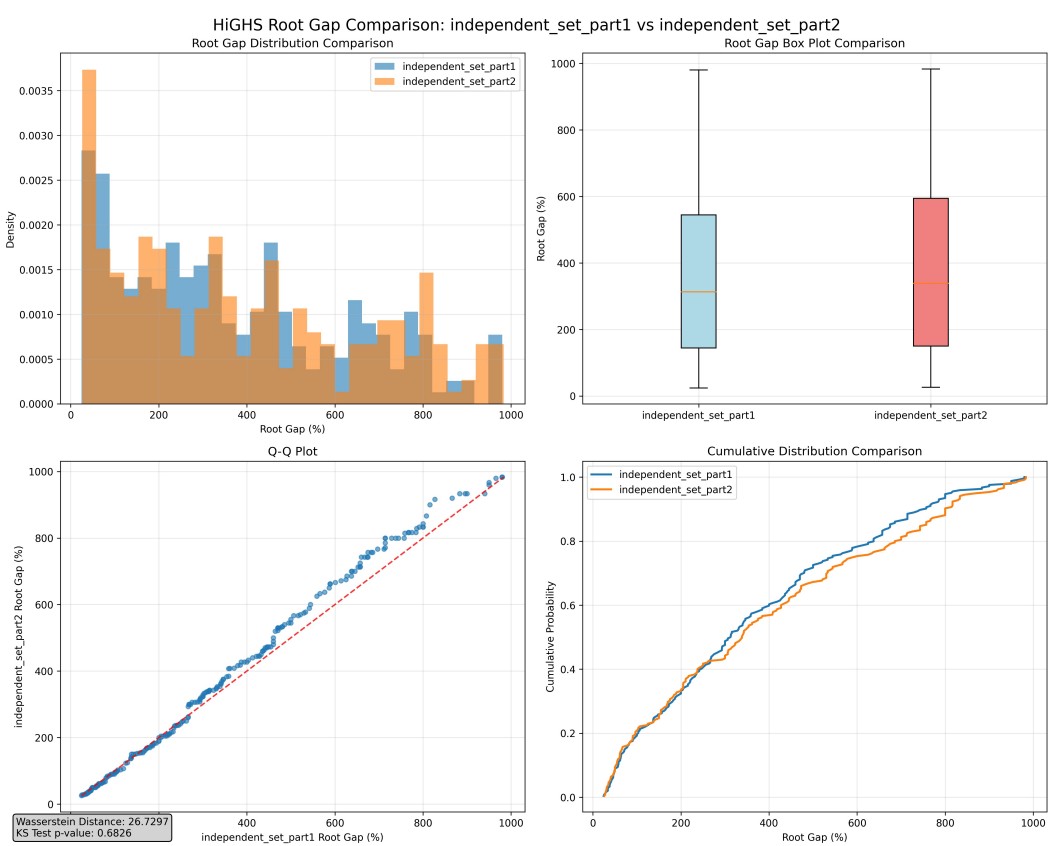

Figure 10: Comparison of root node gap distributions for HiGHS in split-half validation. The two distributions represent the two random halves of the instance set.

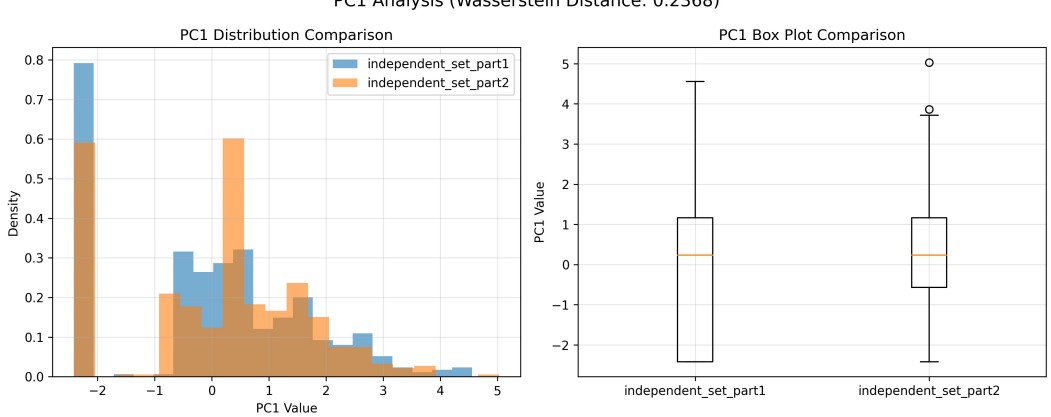

Figure 11: Comparison of heuristic distributions for HiGHS in split-half validation. The two distributions represent the two random halves of the instance set.

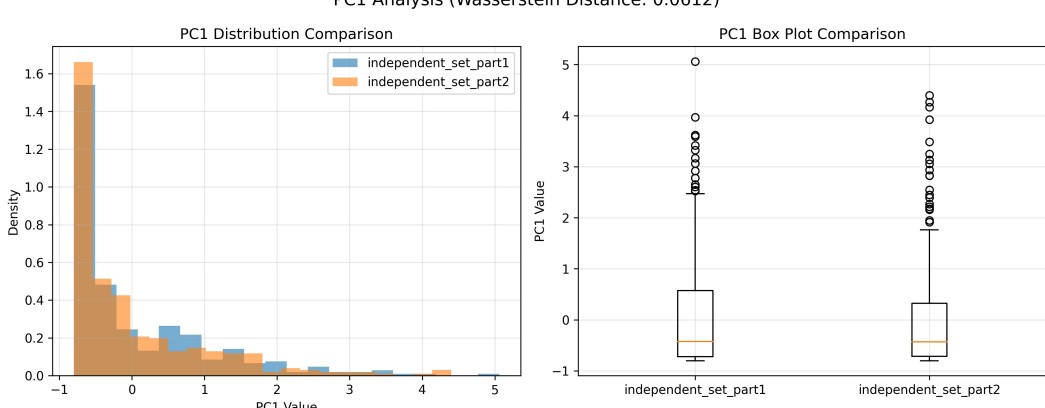

Figure 12: Comparison of cut plane distributions for HiGHS in split-half validation. The two distributions represent the two random halves of the instance set.

a diagnostic tool to analyze and compare the internal stability and dynamic behavior of different solvers.

### L.4 SOME INSIGHTS

We noted a stark divergence in the performance of the same G2MILP model on IS and SC (Set Covering) problems. For SC problems, G2MILP largely replicated the performance characteristics of the original instance set, whereas for IS problems, the difficulty of the generated instances increased dramatically. This could be attributed to the following reasons:

For the IS problem, the objective is to find the largest subset of nodes in a graph such that no two nodes within the subset are adjacent (Tarjan & Trojanowski, 1977; West, 2001). In the MILP formulation, a binary variable $x_i$ is typically assigned to each node $i$ (indicating whether node $i$ is in the independent set), and a constraint $x_i + x_j \leq 1$ is added for each edge $(i, j)$ in the graph. In G2MILP's bipartite graph representation, "constraint nodes" correspond to these edge constraints. Thus, when G2MILP modifies a "constraint node", it is altering a constraint associated with a specific edge in the graph. The "structure and hardness" of IS problems are predominantly derived from the global topological properties of the original graph, rather than merely the presence or absence of individual edge constraints or their coefficients (which are typically 1). G2MILP, by modifying individual or a few edge constraints, may struggle to effectively learn or preserve these critical global graph attributes. It might only be performing local "poking" or "connecting" operations, the global structural impact of which can be drastic and unpredictable. The paper mentions that G2MILP "iteratively corrupts and replaces parts of the original graphs". For a problem like IS, which is sensitive to global structure, such local, iterative replacements might more readily lead to structural deviations.

For the SC problem, constraints directly define the covering requirements (Balas & Ho, 1980). Modifications to constraints by G2MILP directly manipulate the core semantic units of the problem. The "structure and hardness" of SC instances in the training data are primarily manifested in the configuration of these constraints (i.e., which constraints exist and which variables they involve) (Balas & Ho, 1980). Therefore, G2MILP's Variational Autoencoder (VAE) has a higher probability of learning effective patterns within these configurations.

This further underscores that structural similarity, in itself, is not conclusively informative and proves useful only for certain problems. This motivates the need for our proposed, more general evaluation metric.

## M BROADER IMPACT

The introduction of the EVA-MILP framework has several potential broader impacts on the fields of mathematical optimization and machine learning:

**Advancing Research in Combinatorial Optimization:** By providing a standardized and comprehensive methodology for evaluating MILP instance generation, EVA-MILP can foster more rigorous and comparable research. This can accelerate the development of higher-quality synthetic instances, which are crucial for testing, benchmarking, and improving MILP solvers.

**Enhancing Machine Learning for Optimization:** The availability of diverse, well-characterized, and challenging MILP instances is vital for training and validating machine learning models aimed at improving optimization algorithms (e.g., for tasks like branching, node selection, or parameter tuning). EVA-MILP can help ensure that the synthetic data used for these purposes more accurately reflects the complexities of real-world problems, leading to more robust and effective ML-driven optimization techniques.

**Development of More Realistic Benchmarks:** The framework encourages a deeper understanding of instance features beyond superficial structural similarity, pushing for the creation of synthetic benchmarks that better capture the computational hardness and nuanced characteristics of operational problems. This can lead to solvers that are better equipped to handle real-world challenges.

**Facilitating Fairer Comparisons:** EVA-MILP offers a more level playing field for comparing different instance generation techniques, moving beyond limited or inconsistent evaluation practices. This transparency can guide researchers and practitioners in selecting or developing generators best suited for their specific needs.

**Educational Tool:** The framework and its associated metrics can serve as an educational resource for students and researchers new to MILP, providing insights into what constitutes a 'good' or 'hard' instance and how various features influence solver performance.

Overall, the EVA-MILP framework is intended to be a positive contribution, aiming to improve the quality, diversity, and understanding of MILP instances used in research and development. Its primary societal impact is expected to be the indirect advancement of optimization technology, leading to better solutions for complex decision-making problems across various sectors."

## N    LLM USAGE STATEMENT

During the preparation of this manuscript, we employed a Large Language Model to assist with language editing. Its primary role was to correct grammar, refine sentence structures, and improve the overall clarity and readability of the text. The LLM was used solely for linguistic enhancements—such as polishing phrasing and improving logical flow—so that the research is communicated in a clear and professional manner. All scientific ideas, experimental methods, analyses, and conclusions reported in this paper are entirely the authors' own work; the LLM's contribution was limited strictly to language refinement.

