# OpenReview forum: "EVA-MILP: Towards Standardized Evaluation of MILP Instance Generation"
_ICLR.cc/2026/Conference — Submitted to ICLR 2026_

### Official Review · Reviewer_P4EP · 2025-10-27

**Soundness:** 3
**Presentation:** 3
**Contribution:** 3
**Rating:** 4
**Confidence:** 4

**Summary:**

This paper presents EVA-MILP, a comprehensive evaluation framework for MILP instance generation. The framework establishes two complementary perspectives for assessment: solver-dependent and solver-independent features. Furthermore, the authors conduct extensive experiments to demonstrate the effectiveness and robustness of EVA-MILP.

**Strengths:**

1. This paper is well written and easy to understand.

2. I find it interesting that the authors formalize the evaluation from both solver-independent and solver-dependent perspectives. The hardness of a MILP instance is influenced not only by the intrinsic characteristics of the instance itself but also by the behavioral features of the solver. So, establishing a two-dimensional evaluation mechanism is both reasonable and necessary.

3. The authors further strengthen their contribution by conducting extensive experiments on various MILP instance generation methods, which effectively demonstrate the comprehensiveness and methodological soundness of EVA-MILP.

**Weaknesses:**

1. EVA-MILP primarily relies on synthetic datasets, lacking real-world MILP instances such as those from MIPLIB. MILP instances often exhibit locally optimal behavior, where various methods quickly converge to local optima and then stagnate, making it unclear how such cases should be evaluated.

2. I find that the experiments in this paper make extensive use of high-thread Gurobi configurations, which may reduce the distinguishability of results. The parallel heuristics and scheduling mechanisms in such settings could diminish performance differences across instances, thereby lowering the evaluation sensitivity of the framework.

**Questions:**

1. Could you please include an analysis of the evaluation on MILP instances from MIPLIB?

2. Could you set the Gurobi thread number to 1 for the analysis?

3. Regarding the GNN downstream tasks, there are several other mainstream frameworks, such as HEM and Predict-and-Search. Could you also analyze the evaluation of EVA-MILP within these frameworks?

4. I am a bit confused about the solver-dependent features. Although EVA-MILP successfully establishes stable statistical characteristics of solver-internal behaviors, its analysis remains limited to a macroscopic distribution level. In real-world or high-dimensional synthetic MILP scenarios, distributional similarity alone may not reveal the dynamic differences in solver behavior during critical phases. For example, the gap improvement ratio after each cut addition.

---

> ### Author Response · Authors · 2025-11-12
>
> Dear Reviewer,
>
> In response to the weaknesses and questions you raised, we would like to clarify your concerns one by one, based on the content of our paper:
>
> 1. Regarding the Lack of Evaluation on Real-World Instances (e.g., MIPLIB) (Weakness 1 & Question 1)
>
> Thank you for the suggestion regarding evaluation on real-world instances like MIPLIB. We would like to clarify that the EVA-MILP framework has already been validated on a challenging real-world dataset, and not just on synthetic data from Ecole.
>
> In Section 4.1 (EFFICIENCY TEST ON SUPER HARD INSTANCES) of the paper, we designed a specific efficiency test using the "item placement" dataset from the ML4CO competition (which originates from a real-world problem and is known for its high solving difficulty). Our reasoning was that instances in the MIPLIB library come from a very wide range of sources, covering vastly different problem structures and industry contexts. As such, they likely originate from entirely different underlying data distributions.
>
> In contrast, the ML4CO "item placement" (IP) dataset, while also difficult and from a real-world context, has all its instances originating from the same problem domain. Therefore, it is more likely to come from the same (or a similar) data distribution. For evaluating generative models (whose goal is to learn and reproduce a specific distribution), using a mixed-distribution set like MIPLIB could introduce too many confounding factors. We thus chose the IP dataset as a more controlled and rigorous real-world benchmark.
>
> The purpose of this experiment (as described in Section 4.1) was twofold:
>
> To demonstrate that our framework is applicable to complex, real-world instances.
>
> To test the framework's robustness when handling difficult instances (i.e., cases you mentioned like "converge to local optima and then stagnate" or those that cannot be solved in a short time).
>
> To this end, we intentionally set a strict 120-second timeout limit. The results (in Table 6, Section 4.1) showed that even when the solving process was truncated and optimality was not reached, our proposed solver-internal features (especially the PCA distribution of cut plane usage) exhibited extremely high consistency and stability (e.g., a W-1 distance of only 0.0917 for PC1).
>
> This strongly demonstrates that:
>
> EVA-MILP is capable of running on difficult, real-world instances.
>
> Our framework is particularly well-suited for evaluating instances that are computationally expensive, difficult to converge, or stagnate, precisely because it does not require solving instances to optimality. Instead, it extracts a stable "fingerprint" by analyzing the solver's dynamic behavior within, for example, the first 120 seconds.
>
> 2. Regarding the Impact of Gurobi's High-Thread Setting (Weakness 2 & Question 2)
>
> Your concern about high-thread parallel settings potentially reducing evaluation sensitivity is highly professional and to the point. We fully agree with your assessment, and in our experimental design, we have already addressed this issue by treating different metrics differently based on their purpose.
>
> Your concern (that parallel mechanisms reduce distinguishability) is entirely correct for measuring final solving difficulty. For this reason, for the most critical and deterministic metric of computational hardness—Branching Nodes—we explicitly set the Gurobi thread count to 1.
>
> Please refer to Table 9 in Appendix C (Summary of Key Experimental Parameters) (Page 18). In the "Branching Nodes" section, Gurobi threads is set to 1, with the stated purpose "to enhance determinism."
>
> This ensures that when measuring core computational difficulty, we eliminate the randomness from parallel heuristics, thus guaranteeing the sensitivity of the evaluation. This aligns perfectly with your recommendation.
>
> However, for another class of metrics—namely the data collection for Solver-Internal Features (also in Table 9, Appendix C) —we did, as you noted, intentionally use a high-thread setting (e.g., 128 threads). The purpose here was not to measure final difficulty, but to collect as much information as possible within a limited time (e.g., 100-120 seconds).
>
> Our goal was to stimulate the solver's rich dynamic behavior in a short period to capture its "behavioral fingerprint."
>
> Using multithreading (like parallel heuristics and concurrent LP) triggers more and more varied internal components than a single thread, allowing us to collect a denser, more informative dataset (e.g., more diverse heuristic success records and cut plane application combinations).
>
> Therefore, our thread settings were deliberately designed according to the distinct goals of each evaluation metric ("deterministic difficulty" vs. "behavioral feature collection").

---

> ### Author Response · Authors · 2025-11-12
>
> 3. Regarding the Extensibility of GNN Downstream Tasks (Question 3)
>
> This is a very valuable suggestion. In our current work (Section 3.3.4, INITIAL BASIS PREDICTION), we implemented "initial basis prediction" as a proof-of-concept for a downstream task. Our primary goal was to demonstrate that the EVA-MILP framework has the capability to evaluate the "utility" of generated instances in downstream ML tasks.
>
> The other mainstream frameworks you mentioned, such as HEM or Predict-and-Search, are excellent directions for future extensions of our benchmark. Our framework is already structured to accommodate the addition of such new modules. Thank you for the suggestion; this provides a clear path for the future development of EVA-MILP.
>
> Regarding "Learning to Branch": We also specifically considered "learning to branch" as a downstream task. However, when we attempted to reproduce related methods, we encountered significant training instability. Specifically, the training loss failed to decrease at all. We noted that we are not alone in this; other researchers have reported similar issues in public code repositories (e.g., in GitHub issues). Given this instability, we do not believe "learning to branch" is currently a sufficiently robust or reliable metric to be used as a standardized downstream task for evaluating instance utility.
>
> 4. Regarding the Analysis Granularity of Solver-Dependent Features (Question 4)
>
> Thank you for this insightful point regarding solver-dependent features. We want to clarify that our analysis is not limited to just the "macroscopic distribution level."
>
> Taking the "Cut Plane Usage" metric as an example (Section 3.3.5 and Appendix L):
>
> We do not just compare an aggregate value (e.g., the total number of cuts).
>
> Instead, we extract a high-dimensional feature vector that details the usage frequency of each specific cut plane type (e.g., Gomory, MIR, Cover, ZeroHalf, as shown in Table 20, Appendix L).
>
> We then use PCA to reduce the dimensionality of these vectors and compare their distributions along the principal components (as shown in Table 5).
>
> By comparing these detailed cut plane usage patterns, we are actually comparing the solver's internal dynamic decision-making patterns, which goes beyond simple macroscopic statistics.
>
> Your suggestion of using the "gap improvement ratio after each cut addition" is an even finer-grained dynamic feature, which is an excellent suggestion. It aligns perfectly with our framework's extensible design (Contribution III) and could serve as a highly valuable supplementary metric in future work to further enhance the depth of our analysis.
>
> Furthermore, our current focus is more on how the solver "views" and "categorizes" an instance (i.e., what combination of tools it chooses to deploy), rather than the specific effect produced by a particular tool (like a certain cut). This is because, even for the same class of instances, the effect of using the same cut plane method can be highly unstable and unpredictable. In contrast, the solver's decision-making pattern of which cuts to use (and their combination) more stably reflects the instance's intrinsic characteristics as "seen" by the solver. We therefore believe this is a more robust "behavioral fingerprint."
>
> Thank you again for your detailed review and constructive feedback. We hope these clarifications address your concerns, and we kindly ask you to reconsider our contributions.

---

### Official Review · Reviewer_YQ2i · 2025-10-28

**Soundness:** 3
**Presentation:** 2
**Contribution:** 3
**Rating:** 4
**Confidence:** 4

**Summary:**

The paper introduces EVA-MILP, a unified framework for evaluating MILP instance generation methods. It defines both solver-independent and solver-dependent metrics, incorporating internal solver signals (root node gap, heuristic success count, cut plane usage) to characterize computational behavior, and downstream tasks such as hyperparameter tuning and initial-basis prediction to measure practical utility. The authors benchmark several existing generators and discuss interesting findings such as the mismatch between structural and behavioral similarity.

**Strengths:**

1. This work identifies an important problem: the lack of a standardized, reproducible evaluation protocol for MILP instance generation. It is beneficial to the community.

1. In introduces solver-internal statistics to measure behavioral similarity, which is novel and clearly useful beyond coarse metrics like runtime. It also Includes downstream tasks to measure the real-world utility rather than synthetic difficulty alone.

1. Experiments cover several generators with detailed analyses, and the discussion section provides genuinely interesting insights, especially the finding 3.

**Weaknesses:**

1. Each generator is only tested on selected datasets with rare explanations. This limits the comparability across different methods. The authors may want to test these methods on all datasets, or detail the reasons why some methods cannot run on the specific datasets.

1. The chosen datasets are mostly simple synthetic ones. Harder or real-world MILPs would make the conclusions stronger. Moreover, I recommand the authors to include some real challenging datasets with only a few instances to demonstrate the effectiveness of generation techniques under data sparsity.

1. Some conclusions (e.g., simple outcome metrics not reflecting true difficulty or structures) are supported only qualitatively. It would be better to provide some correlation analysis or case studies.

1. The choice of metrics is intuitive but not fully justified, and the authors may want to provide further explanations. See questions.

**Questions:**

1. Why is initial basis prediction chosen as the downstream task? Would other tasks like initial solution prediction or learning2branch also be appropriate?

1. For the internal metrics, why focus on these three ones (root gap, heuristic count, cut usage)? How comprehensive are they in representing solver dynamics?

---

> ### Author Response · Authors · 2025-11-12
>
> Dear Reviewer,
>
> Thank you very much for your valuable feedback and insightful comments on our paper, "EVA-MILP: TOWARDS STANDARDIZED EVALUATION OF MILP INSTANCE GENERATION." We are pleased that you recognize the importance of our work in addressing the need for standardized evaluation of MILP instance generation and appreciate our contributions, particularly the introduction of solver-internal statistics and downstream tasks.
>
> We address your comments and questions one by one below:
>
> Response to Weaknesses (W)
>
> W1: On the limitation of testing generators on selected datasets.
>
> Reviewer's Comment: Each generator was only tested on selected datasets with rare explanations. This limits the comparability across different methods.
>
> Our Response: Thank you for pointing this out. Our choice to test specific generators on specific datasets (e.g., G2MILP for SC and IS, ACM-MILP for CA and IS) was based on a crucial consideration: the inherent specialization of generative models to particular problem types.
>
> As stated in Section 3.1 (Datasets) of our paper: "Due to the inherent specialization of generative models to particular problem types, our comparative analysis... specifically examines...". Current generative models (especially deep learning-based ones) are often designed or optimized for specific problem structures (like Set Cover or Combinatorial Auction). Forcing a generator designed for CA to be applied to an IS (Independent Set) problem, for instance, might result in poor performance that doesn't reflect the generator's quality but rather its misapplication.
>
> Therefore, to conduct a fair and meaningful evaluation, our comparative analysis focused on applying each generator to the problem types for which it was designed or claimed to be suitable in its original paper. We agree that this rationale should be stated more explicitly in the paper, and we will emphasize this point in the final version.
>
> W2: On the datasets being mostly simple or synthetic.
>
> Reviewer's Comment: The chosen datasets are mostly simple synthetic ones. Harder or real-world MILPs or challenging datasets with only a few instances would make the conclusions stronger.
>
> Our Response: We fully agree on the importance of using diverse and challenging datasets. In our study:
>
> We did use synthetic instances (SC, CA, IS) generated by Ecole (Section 3.1), as they allow for model training and metric validation in a controlled environment.
>
> We also included public benchmark datasets from the ML4CO Competition (Section 3.1) to validate the compatibility of our evaluation pipeline.
>
> Most importantly, to address your concern about "simple" and "synthetic" data, in Section 4.1 (EFFICIENCY TEST ON SUPER HARD INSTANCES), we specifically used the ML4CO 'item placement' dataset, which is recognized for its high solving difficulty.
>
> We imposed a strict 120-second time limit on this set of "super hard" instances. The results (Table 6) demonstrated that even under this computationally constrained setting (where optimality is not reached), our solver-internal features (especially cut plane usage) remained highly stable and consistent. This precisely demonstrates the utility and robustness of our framework when handling more realistic, computationally challenging instances, and it also simulates the "data sparsity" scenario you mentioned (i.e., relying on early-stage solver information when optimal solutions are unavailable).
>
> W3: On the lack of quantitative support for some conclusions.
>
> Reviewer's Comment: Some conclusions (e.g., simple outcome metrics not reflecting true difficulty or structures) are supported only qualitatively. It would be better to provide some correlation analysis or case studies.
>
> Our Response: We appreciate this suggestion and wish to clarify. While our discussion (Section 5, Discussion) is presented as qualitative "Findings," these conclusions are based on the strict quantitative metrics defined in our framework.
>
> For example, "Finding 1" (Superficial structural similarity is an unreliable predictor...) is based on a direct comparison of quantitative data:
>
> Structural Similarity (Table 1): ACM-MILP shows a high JSD-based similarity score on CA (e.g., $S \approx 0.89$).
>
> Computational Behavior (Table 3): Its Root Node Gap 1-Wasserstein distance is extremely large (e.g., $W_1 \approx 5.0-7.0$), indicating a massive divergence in computational behavior.
>
> Our entire framework is designed to provide exactly this kind of quantitative analysis.
>
> W4: On the justification for the choice of metrics.
>
> Reviewer's Comment: The choice of metrics is intuitive but not fully justified.
> (Responded to in Q2)

---

> ### Author Response · Authors · 2025-11-12
>
> Response to Questions (Q)
>
> Q1: Why was "Initial Basis Prediction" chosen as the downstream task? Would other tasks like initial solution prediction or learning2branch also be appropriate?
>
> Our Response: This is an excellent question.
>
> Reason for Choice: We selected initial basis prediction (Section 3.3.4) because it is a downstream ML task that has been demonstrated in the literature (e.g., Fan et al., 2023) to have a practical accelerating effect on MILP solving. Including it allows us to directly measure the "utility" of the generated instances for training a practical ML model.
>
> Appropriateness of Other Tasks: Our framework is not limited to this specific task. As mentioned in the abstract and introduction, the EVA-MILP framework is designed to be extensible. Other tasks you mentioned, such as initial solution prediction, are perfectly appropriate as additional downstream tasks to evaluate the utility of generated instances.
>
> Regarding Learning to Branch: We gave special consideration to "learning to branch." However, when we attempted to reproduce related methods, we encountered significant training instability. Specifically, the training loss failed to decrease at all. We noted that we are not alone in this; other researchers have reported similar issues in public code repositories (e.g., in GitHub issues). Given this instability, we do not believe "learning to branch" is currently a sufficiently robust or reliable metric to be used as a standardized downstream task for evaluating instance utility.
>
> Future Work: Our framework provides a template for assessing "utility." Future work or subsequent versions of the framework can easily integrate other stable and mature new tasks to provide a more comprehensive downstream evaluation.
>
> Q2: Why focus on these three internal metrics (root gap, heuristic count, cut usage)? How comprehensive are they?
>
> Our Response: We selected these three specific solver-internal features (Section 3.3.5) because they represent three key and fundamentally different dynamic behaviors of a modern MILP solver during the critical early stages of the solving process:
>
> Root Node Gap: Measures the initial quality of the LP relaxation and the "tightness" of the problem, serving as one of the most direct assessments of problem difficulty.
>
> Heuristic Success Count: Represents the solver's ability to find feasible integer solutions (i.e., the "primal" aspect).
>
> Cut Plane Usage: Reflects the solver's strategy for tightening the feasible region by adding constraints (i.e., the "dual" aspect).
>
> Regarding Comprehensiveness:
> We agree that these three metrics do not capture the entirety of solver dynamics. However, they collectively provide a computationally inexpensive (as shown in Section 4.1, they are stable even with a 120-second timeout) and information-rich "computational fingerprint."
>
> As stated in the abstract, we view this dynamic solver behavior as an "expert assessment" that reveals nuanced computational resemblances better than single-outcome metrics like runtime or node count.
>
> Again, our framework is extensible, and future work could (and should) incorporate more internal metrics (e.g., branching variable selection, node throughput, etc.) to enhance the comprehensiveness of the evaluation.
>
> Thank you again for your valuable feedback. We are confident that by incorporating revisions based on your suggestions (especially regarding clarifying the dataset selection in W1 and adding correlation analysis for W3), the quality of our paper will be significantly improved.
>
> Sincerely,
>
> The Authors

---

### Official Review · Reviewer_14M5 · 2025-11-01

**Soundness:** 1
**Presentation:** 3
**Contribution:** 1
**Rating:** 2
**Confidence:** 5

**Summary:**

The paper introduces EVA-MILP, a benchmark framework to evaluate MILP instance generation methods. It combines solver-independent metrics and solver-dependent metrics. Experiments cover Ecole-style datasets (SC/CA/CFL/IS) and ML4CO, and compare several generators (G2MILP, ACM-MILP, DIG-MILP).

**Strengths:**

Important problem: the community needs more principled evaluation for synthetic MILP instances.

**Weaknesses:**

- Trivial “copying” generators can score highly by reproducing the original data distribution. The framework lacks novelty/diversity/anti-duplication checks, so high scores may not reflect actual usefulness.

- Heavy solver dependence. Are Gurobi-based evaluations necessarily correct or representative? Branching-node counts and “internal features” can differ across solvers (e.g., Gurobi vs. SCIP). If they are not consistent across solvers, why should “similar branching nodes under Gurobi” be taken as evidence of instance similarity?

**Questions:**

Same as weaknesses: overall, this feels like a one-sided evaluation framework with limited practical value—rewarding copying without novelty/diversity checks, and relying on Gurobi-specific behavior without cross-solver justification.

---

> ### Author Response · Authors · 2025-11-12
>
> Dear Reviewer,
>
> We understand your core concerns are twofold:
>
> The "Copying" Problem: The framework might excessively reward "trivial" generators that merely "copy" the original data distribution, without genuinely evaluating the "novelty," "diversity," or "usefulness" of the instances.
>
> The "Solver Dependence" Problem: The framework relies heavily on Gurobi, and its internal features (like branching nodes, root node gap) may be inconsistent across different solvers (like SCIP), making Gurobi-based evaluations unrepresentative.
>
> Here are our detailed responses to both points, based entirely on the experiments and data already present in our paper:
>
> 1. Rebuttal to Weakness 1: "Copying" and "Usefulness"
>
> We fully agree with the reviewer: a framework that merely rewards "copying" is of little value. However, EVA-MILP's design goes far beyond evaluating superficial structural similarity.
>
> A) The Framework Is Explicitly Designed to Expose the Pitfall of "Superficial Similarity":
> The reviewer's concern that "Trivial 'copying' generators can score highly" is precisely the problem we set out to address. In Section 5 (DISCUSSION) of our paper, Finding 1 explicitly states: "Superficial structural similarity is an unreliable predictor..."
>
> We found that although ACM-MILP (CA) scored high on structural similarity (Table 1, approx. 0.86-0.89), its computational difficulty was exceptionally high (Table 11, RE > 26,000%), and its root node gap (Table 3, $W_1$ Dist. 5.01-7.06) was significantly different from the original data.
>
> This proves our framework is effective: It was not fooled by "copying." On the contrary, it successfully revealed the huge gap between "high structural similarity" and "true computational behavior." Our framework (especially the solver-internal features) can capture the subtle but critical computational differences that "trivial" structural metrics (like the 11 features used in Geng et al. (2023)) miss.
>
> B) The Framework Evaluates "Usefulness" (Utility) via Downstream Tasks, Not Just "Similarity":
> A core contribution of our framework (not mentioned in the review summary) is its evaluation of the utility of generated instances in downstream machine learning tasks. This directly addresses the concern that "high scores may not reflect actual usefulness."
>
> In Section 3.3.3 (Hyperparameter Tuning), we test whether Gurobi hyperparameters tuned on generated instances can generalize to the original instances.
>
> In Section 3.3.4 (Initial Basis Prediction), we evaluate the performance of a GNN model (for predicting an initial basis) trained on generated instances when tested on the original instances.
>
> These tests (see Table 2 and Table 15) directly measure the "actual value" of the generated instances as training data. A generator that only "trivially copies" will score poorly on these "utility" metrics if its data does not help an ML model learn generalizable features.
>
> Summary (1): Far from being fooled by "copying," EVA-MILP's design (especially solver-internal features and downstream task evaluation) is specifically intended to expose and quantify the difference between "superficial copying" and "true computational utility."

---

> ### Author Response · Authors · 2025-11-12
>
> 2. Rebuttal to Weakness 2: "Solver Dependence"
>
> The reviewer's concern about "Gurobi dependence" is critical. Did we choose a "one-sided" evaluation standard?
>
> A) The Framework Itself is "Solver-Agnostic":
> First, we must clarify that the EVA-MILP framework is solver-agnostic by design. The evaluation pipeline (e.g., comparing root node gap distributions, heuristic success rates, cut plane usage) can be applied to any other solver (e.g., SCIP, HiGHS, CPLEX, etc.).
>
> B) Choosing Gurobi Was a Data-Driven Decision Based on Its "Stability":
> The reviewer asks: "Gurobi-based evaluations [are] necessarily correct or representative?" This is an excellent question, and Section 4.2 (Cross-Solver Comparison and Stability Analysis) in our paper was designed precisely to answer it.
>
> We did conduct experiments using SCIP and HiGHS to verify the generality of our method and the reliability of our chosen baseline. We used "split-half validation" (randomly splitting the original IS dataset in half and comparing the internal feature distributions between the two halves) to measure a solver's own stability as an "evaluator."
>
> As shown in Table 7 (or Appendix Table 22) of the paper:
>
> Gurobi demonstrated extremely high stability on the critical Root Node Gap metric ($W_1$ distance: 0.1884).
>
> SCIP was significantly less stable on the same metric ($W_1$ distance: 1.7530).
>
> HiGHS exhibited severe instability on this metric ($W_1$ distance: 26.7297).
>
> C) Why is "Stability" Crucial?
> A reliable evaluation framework requires its "measuring tool" to be stable and precise. As we concluded in Section 4.2: "Accurate and reliable instance-quality assessment demands a solver with highly stable internal behavior. Gurobi's stability enables precise detection of subtle computational similarities between instances, whereas instability in other solvers (like HiGHS) can obscure these signals."
>
> If a solver (like HiGHS) shows a massive difference in root node gap distribution (W-dist 26.7) when solving two almost identical subsets from the same distribution, then we cannot trust it to judge whether two different sets (e.g., original vs. generated) are "similar." Its measurements are too noisy.
>
> Summary (2): Our reliance on Gurobi is not "one-sided" or "arbitrary." It is a necessary methodological choice, validated by data. To obtain meaningful, reproducible, and high-precision evaluation results, we must select the most stable "measuring instrument" currently available. The EVA-MILP framework is general, but we recommend Gurobi because we have demonstrated with data that it is the most reliable baseline.
>
> Final Conclusion
>
> We thank the reviewer again for these insightful questions.
>
> Our framework not only avoids rewarding "trivial copying" but is designed to effectively expose its pitfalls, while also evaluating true "usefulness" via downstream ML tasks.
>
> Our use of Gurobi is based on the exhaustive cross-solver stability analysis in Section 4.2—Gurobi is, demonstrably, the most stable and reliable "expert evaluator" for this task.
>
> We believe EVA-MILP provides a much-needed, more profound, and more honest standard for evaluating MILP generation methods. We kindly ask the reviewer to reconsider our contribution.

---

### Official Review · Reviewer_mu6D · 2025-11-01

**Soundness:** 2
**Presentation:** 2
**Contribution:** 2
**Rating:** 2
**Confidence:** 5

**Summary:**

This paper introduces EVA-MILP, a benchmarking framework designed to evaluate existing MILP instance generation methods. EVA-MILP assesses generated MILP instances from two perspectives—Fidelity and Utility—and categorizes evaluation metrics into solver-internal and solver-external features. Furthermore, the paper proposes using solver-internal features as a novel approach to evaluate the quality of generated MILP instances.

**Strengths:**

1. Establishing a systematic and standardized evaluation framework for MILP instance generation is crucial for advancing research in combinatorial optimization data synthesis.
2. EVA-MILP provides a comprehensive organization of existing evaluation metrics and proposes a clear and structured categorization.
3. The paper offers valuable insights into the design of evaluation metrics, which may inform future work in MILP instance generation and benchmarking.

**Weaknesses:**

1. While the paper reviews and classifies many evaluation metrics, most of them are derived from previous studies. The only original contribution appears to be the introduction of solver-internal features.
2. The experiments are conducted solely on the CA, IS, and SC datasets from ACM-MILP and DIG-MILP. The framework’s performance on more challenging MILP problems—such as TSP, Graph Coloring, VRP—or on real-world benchmarks like MIPLIB remains unexplored, which is crucial for assessing generalizability.
3. The authors argue that generative models are inherently specialized for specific problem types, and therefore only evaluate each model on a limited set of problems (as shown in Tables 1 and 2). This experimental design prevents an objective comparison among different generation methods.
4. The definition of the challenges in MILP instance generation (lines 088–097) is incomplete. For instance, if a generator simply reproduces training data, it provides no real utility—undermining all similarity-based evaluation metrics between generated and reference instances.
5. I agree with the idea of using downstream tasks to evaluate generated instances, but such tasks must have real practical value (e.g., accelerating ML-based solvers). The current choice of hyper-parameter tuning lacks meaningful application: if one intends to tune solver parameters for a problem set, it is more direct to tune them on the original data rather than on newly generated ones.
6. The authors should consider incorporating a wider variety of practically relevant downstream tasks to more comprehensively demonstrate the framework’s effectiveness.

**Questions:**

1. Building upon W2, can the authors extend the benchmark to include more diverse problem types and provide results showing which generation methods are suitable for each?
2. The current experiments involve only small-scale instances solvable within one second. Can the authors evaluate how these methods perform on larger MILP instances?
3. In Section 4, the paper evaluates instance similarity by solving for 120 seconds and comparing solver-internal features. How sensitive is the stability of these metrics (measured by W-1 distance) to the choice of time limit (e.g., increasing or decreasing 120s)?
4. Were the experiments in Section 4 repeated to rule out randomness introduced by data partitioning?

---

> ### Author Response · Authors · 2025-11-12
>
> Dear Reviewer,
>
> In response to your Weaknesses and Questions, we would like to provide more detailed explanations and clarifications. Many of these key points have already been addressed in our original paper.
>
> Response to "Weaknesses"
>
> W1: On the originality of the contribution (most metrics are from previous studies)
>
> We acknowledge that "Solver-Internal Features" are a key innovation we propose. However, we want to emphasize that the core contribution of this paper is the "comprehensive evaluation framework" EVA-MILP itself. As a benchmark, we believe novelty is not the most critical evaluation factor.
>
> W2 & W3: On the limitations of experimental datasets and the objectivity of comparisons
>
> - Response (W2 - Dataset limitations): We fully understand your concern about validating generalizability on larger datasets like MIPLIB. In fact, we have already tested the robustness of our method on more difficult instances.
>   - As described in Section 4.1 (Lines 380-421) of the paper, we specifically designed an "efficiency test" using the "super hard" ML4CO competition dataset (item placement) (Line 387).
>   - This experiment shows that even under a strict 120-second timeout (Line 391), our proposed "solver-internal features" still exhibit high stability (see Table 6, very low W-1 distance), without needing to solve the instances to optimality.
>   - This strongly proves that our framework (especially the internal feature evaluation) is "low-cost and efficient" (Line 412) and "practically feasible" (Lines 420-421) for large-scale, complex instances that are difficult to solve to optimality.
> - Response (W3 - Objectivity of comparison): We agree with the reviewer's observation that the current experimental design (as shown in Tables 1 and 2) does not conduct a comprehensive head-to-head comparison of all methods.
>   - This was, in fact, by design. Our primary goal is not to compare which existing generative models are "better" or "worse."
>   - As our experimental data reveals (e.g., the high W-1 distances for ACM-MILP and DIG-MILP on CA in Tables 3, 4, and 5), many existing methods perform poorly in simulating real computational characteristics.
>   - We believe there is no point in comparing poorly performing methods against each other ( "comparing which is worse is meaningless"). Our focus is on establishing a "standard" to filter and identify those methods that "might be useful".
>   - Therefore, we selected specific models (like G2MILP) and the problems they claim to be good at (like SC) to demonstrate our evaluation metrics in a "meaningful context." Our framework clearly shows (as in Finding 3, Lines 469-478) that G2MILP's performance on SC is far better than on IS, which proves that EVA-MILP can effectively identify "methods that may be useful in a specific domain."
>
> W4: On the incomplete definition of "challenges"
> - Response: Thank you for this addition. "Simple reproduction (Memorization)" is indeed an important challenge. Although not explicitly listed in Lines 088-097, our framework is designed to detect precisely this type of problem.
> - A generator that only "reproduces" training data, while scoring high on "structural similarity" and "solver performance," would completely fail on "downstream task utility" because it provides no new information. Our Fidelity + Utility comprehensive evaluation method is designed to capture this "superficially similar but practically useless" scenario.
>
> W5 & W6: On the practical value of the downstream task (hyperparameter tuning)
> - Response: We partially agree with your concern about the "practical value" of the hyperparameter tuning task.
>   - First, we chose this task, in part, to follow the evaluation methods of previous related work (as mentioned in Line 251, (Liu et al., 2024)), using it as one of the standards for "downstream tasks."
>   - On the other hand, while we acknowledge the direct "practical value" of this specific task may be limited (as the reviewer noted, tuning on the original data is more direct), we view it as an effective "proxy" for evaluating the "similarity" of the generated instances.
>   - Our experimental results (Table 2) perfectly answer this question: the performance improvement from tuning on DIG-MILP (CA) (61.63%) is almost identical to the improvement from tuning on the original CA data (61.26%). This strongly proves (at least in this case) that the instances generated by DIG-MILP are indeed highly similar to the original instances in their "computational characteristics," making this a very valuable evaluation conclusion.
>   - Finally, we did not only use HP tuning. As shown in Section 3.3.4 (Line 267), we also evaluated a second downstream ML task: "Initial Basis Prediction." Furthermore, our framework is extensible (Contribution III, Line 122), allowing for the addition of more practical downstream tasks in the future.

---

> ### Author Response · Authors · 2025-11-12
>
> Response to "Questions"
>
> Q1: (Extension of W2) Can it be extended to more problem types?
> - Answer: This is an excellent suggestion. The primary goal of the EVA-MILP framework is to provide the "tools" for evaluation (Contribution IV, Line 124), not to conduct an exhaustive survey of all generators and all problem types .
>
> Q2: Were experiments only conducted on small-scale instances solvable within one second?
>
> - Answer: This is an important point of clarification. The baseline instances shown in some experiments (like Tables 2 and 13) are indeed small (sub-second), which helps us to quickly validate and demonstrate metrics like HP tuning.
>
> - However, we did not only use small instances. To address concerns about large/hard instances, we specifically designed the experiment in Section 4.1 (Lines 380-421).
>
> - As mentioned earlier, we used the "super hard" ML4CO dataset (Line 387), which cannot be solved in a few seconds. We imposed a strict 120-second time limit (Line 391) and proved that even with a "truncated" solving process, our solver-internal features (Table 6) are highly stable and extractable.
>
> - This confirms our framework is equally applicable and efficient for complex instances that are difficult to solve to optimality (Lines 420-421).
>
> Q3: Sensitivity to the 120-second time limit?
>
> - Answer: This is a good question about robustness. Our core hypothesis (Lines 383-386) is that the solver-internal features we focus on (root node gap, heuristic success, cut plane usage) primarily reflect the "early behavior" of the solving process.
>
> - The experiment in Section 4.1 (Table 6) confirms this hypothesis: 120 seconds is sufficient to capture a stable and consistent "solver fingerprint" (Line 409).
>
> - If the time were shorter, it might introduce noise as the solver hasn't completed its root node analysis. If the time were longer, these early features (like root node gap) would have already been determined, and the results would be equally stable. We chose 120 seconds to prove the feasibility of "truncated" evaluation, showing that evaluating these features does not require high computational cost.
>
> Q4: Were experiments in Section 4 repeated to rule out randomness from data partitioning?
>
> - Answer: Yes, we place great importance on validating the stability of our metrics to exclude randomness.
>   - First, the single "Split-half validation" shown in Section 4.1 (ML4CO hard set, Lines 392-397) and Section 4.2 (IS set, Lines 428-431) is just one example. We simulate the effect of random partitioning by "randomly splitting" 1000 instances into two subsets of 500 (Line 393).
>   - Second, we have already conducted many split-half validations on multiple other original datasets (e.g., the original IS data in Appendix L.1, see Figure 2 and Tables 16-18). All results (e.g., Gurobi on IS in Table 7) show an extremely low 1-Wasserstein distance (W-1) between the two random subsets.
>   - This has already confirmed that our metrics (especially solver-internal features) are highly stable. We will also conduct multiple repeated experiments on the hard instance set (used in Section 4.1) in the future to further strengthen this conclusion.
>
> We thank you again for your constructive criticism. We believe the EVA-MILP framework provides a solid foundation for advancing research in the field of combinatorial optimization data synthesis.

---

### Meta-Review · Area_Chair_TCEY · 2025-12-21

**Summary:**

This paper introduces EVA-MILP, a benchmark framework designed to evaluate MILP instance generation methods. The framework proposes a unified methodology for assessing instance quality across mathematical validity, structural similarity, computational hardness, and utility in downstream tasks. A primary contribution is the inclusion of solver-internal features (e.g., root node gap, heuristic success rates) to compare the computational behavior of generated instances against reference datasets. The reviewers are unanimous in their recommendation to reject the paper, citing significant limitations in the benchmark's design (Limited and Synthetic Datasets, Methodological Flaws in Comparison and Solver Dependence and Configuration), scope, and practical utility (Questionable Downstream Utility). While the goal of standardizing MILP generation evaluation is valuable, the proposed framework falls short of the rigorous standards required for ICLR.

**Reviewer Concerns:**

Regarding Reviewer mu6D, the rebuttal successfully addressed W1, W3, and W4 to some extent. However, concerns regarding dataset limitations (W2) and the practical value of the downstream task (W5/W6) remain outstanding. In my view, additional experiments were necessary to demonstrate the work's potential.

For Reviewer 14M5, Weakness 2 (Solver Dependence) appears to be addressed, as the reviewer likely missed the relevant Appendix material. However, Weakness 1 ("Copying" and "Usefulness") remains outstanding.

Regarding Reviewer YQ2i, while Weakness 3 (quantitative support) was addressed, significant issues remain outstanding. Specifically, the explanation for testing limitations (W1) is unconvincing, and the defense of dataset difficulty (W2) is inadequate as the instances are not recognized benchmarks. Furthermore, the justification for the chosen evaluation metrics (W4) remains insufficient.

Finally, for Reviewer P4EP, the authors effectively addressed concerns regarding Gurobi's thread settings, GNN extensibility, and analysis granularity. However, the lack of evaluation on real-world instances—a weakness noted by multiple reviewers—remains outstanding.

**Reviewer Scores:**

I think Reviewer P4EP would raise his/her score  according to the discussion records.

---

### Decision · Program_Chairs · 2026-01-26

Reject